# Neuronal lysosomal dysfunction releases exosomes harboring APP C-terminal fragments and unique lipid signatures

André M. Miranda[1,2,3,4], Zofia M. Lasiecka[1], Yimeng Xu[1], Jessi Neufeld[2,5], Sanjid Shahriar[1], Sabrina Simoes[2,5], Robin B. Chan[1,2], Tiago Gil Oliveira[3,4], Scott A. Small[2,5] & Gilbert Di Paolo[1,2,6]

Defects in endolysosomal and autophagic functions are increasingly viewed as key pathological features of neurodegenerative disorders. A master regulator of these functions is phosphatidylinositol-3-phosphate (PI3P), a phospholipid synthesized primarily by class III PI 3-kinase Vps34. Here we report that disruption of neuronal Vps34 function in vitro and in vivo impairs autophagy, lysosomal degradation as well as lipid metabolism, causing endolysosomal membrane damage. PI3P deficiency also promotes secretion of unique exosomes enriched for undigested lysosomal substrates, including amyloid precursor protein C-terminal fragments (APP-CTFs), specific sphingolipids, and the phospholipid bis(monoacylglycero)phosphate (BMP), which normally resides in the internal vesicles of endolysosomes. Secretion of these exosomes requires neutral sphingomyelinase 2 and sphingolipid synthesis. Our results reveal a homeostatic response counteracting lysosomal dysfunction via secretion of atypical exosomes eliminating lysosomal waste and define exosomal APP-CTFs and BMP as candidate biomarkers for endolysosomal dysfunction associated with neurodegenerative disorders.

[1] Department of Pathology and Cell Biology, Columbia University Medical Center, New York, NY 10032, USA. [2] Taub Institute for Research on Alzheimer's disease and the Aging Brain, Columbia University Medical Center, New York, NY 10032, USA. [3] Life and Health Sciences Research Institute (ICVS), School of Medicine, University of Minho, Braga 4710-057, Portugal. [4] ICVS/3B's, PT Government Associate Laboratory, Braga/Guimarães, 4710-057, Portugal. [5] Departments of Neurology, Columbia University Medical Center, New York, NY 10032, USA. [6] Present address: Denali Therapeutics, South San Francisco, CA 94080, USA. Correspondence and requests for materials should be addressed to G.D.P. (email: gd2175@cumc.columbia.edu or dipaolo@dnli.com)

A variety of neurodegenerative disorders are associated with major defects in the endolysosomal pathway. These include Alzheimer's disease[1,2] (AD), Parkinson's disease[3] (PD), frontotemporal dementia (FTD)[4], and several lysosome storage disorders (LSD)[1,5]. The causal relationship between endolysosomal dysfunction and neurodegeneration is demonstrated by the occurrence of rare, LSD-causing familial mutations that affect genes encoding key endolysosomal proteins, typically

**Fig. 1** Vps34 inhibition induces early endosomal abnormalities and causes accumulation of APP-CTFs. **a, b** Representative confocal images of cultured cortical neurons treated with vehicle or VPS34IN1 at 3 μM for 3 h. **a** Super-resolution Airyscan insets and arrows highlight EEA1 endosomes. Bar graphs indicate average EEA1-puncta size, per cell (mean ± SEM, $N = 37$ and 44 cells, respectively, from three independent experiments) and average EEA1/Rab5 puncta intensity, per cell (mean ± SEM, $N = 55$ and 63 for vehicle and VPS34IN1, respectively). **b** Airyscan insets highlight colocalization between APPL1 and Rab5. Bar graph indicates fraction of APPL1 signal colocalizing with Rab5 (mean ± SEM, $N = 35$ cells, from three independent experiments). Scale bar, 10 μm. ***$p < 0.001$ in two-tailed Student's t test. **c** Western blot analysis of Cathepsin D levels in primary cortical neurons treated with vehicle or VPS34IN1 at 3 μM for 24 h. Bar graph represents average protein levels of total CatD (sum of CatD and proCatD) or ratio of CatD/proCatD, normalized to vehicle (mean ± SEM, $N = 7$, from two independent experiments). Right panel, representative confocal images of primary cortical neurons. Airyscan insets highlight luminal sorting of Cathepsin D in LAMP-1-positive compartments. Scale bar, 10 μm. ***$p < 0.001$ in two-tailed Student's t test. **d, e** Western blot analysis of APP metabolites from primary cortical neurons treated as in **c** or N2a cells treated with VPS34IN1 at 1 μM for 24 h. Bar graph denotes average protein levels normalized to vehicle (mean ± SEM, $N = 6$ and 7, respectively, for primary neurons; $N = 8$ and 7, respectively, for N2a cells, from two independent experiments) and murine Aβ40 and Aβ42 levels measured by MSD from culture media (mean ± SEM, $N = 12$ and 13, respectively, for primary neurons; $N = 8$ and 7, respectively, for N2a cells, from two independent experiments). Aβ levels were normalized to lysate total protein. *$p < 0.05$, **$p < 0.01$, ***$p < 0.001$ in two-tailed Student's t test

resulting in aggressive, childhood-onset LSDs associated with neurodegeneration (e.g., Niemann–Pick disease type C or NPC)[5,6]. More recently, mutations in a variety of genes involved in endocytic or endolysosomal function have been shown to cause PD (e.g., *LRRK2*) or some forms of parkinsonism (e.g., *VPS35, SYNJ1, ATP13A2*)[3]. Importantly, homozygous mutations in certain genes cause LSDs (e.g., *GBA* and *GRN*), while heterozygous mutations of the same genes are major genetic risk factors for late-onset neurodegenerative diseases, such as PD and FTD, respectively[3,4]. Finally, several genes involved in the regulation of endolysosomal function have also been linked to late-onset AD (LOAD) suggesting this process to be a key driver of disease onset and progression[7].

Lysosomes are central organelles for the degradation of a large number of macromolecules targeted to this compartment via the endocytic and autophagic pathways[1,5,6]. Failure to properly degrade these substrates leads to their accumulation, within the endolysosomal compartment itself (e.g., glycosphingolipids), in the cytoplasm (e.g., tau, α-synuclein, ubiquitin inclusions) or in the extracellular environment (e.g., amyloid plaques)[1,5,6]. A key molecular pathway controlling endolysosomal function and autophagy is the class III phosphatidylinositol-3-kinase (PI3K-III)/Vps34 signaling pathway, which leads to the phosphorylation of PI on the 3' position of the inositol ring, generating PI3P. Vps34 interacts with p150/Vps15 and Beclin 1, forming either complex I with ATG14L or complex II with UVRAG. Although complex I synthesizes PI3P on pre-autophagosomal membranes, complex II produces PI3P on early endosomes[8,9]. PI3P, in turn, controls the membrane recruitment of a variety of cytosolic effectors harboring PI3P-binding domains, including FYVE and PX modules[9,10]. PI3P also serves as a substrate for the synthesis of $PI(3,5)P_2$ by the PI3P 5-kinase PIKfyve on late endosomes, controlling additional aspects of endolysosomal function[10,11]. Altogether, this pathway mediates a variety of critical processes, such as endosomal fusion, intraluminal vesicle (ILV) budding, endosomal motility as well as the biogenesis and maturation of autophagosomes during macroautophagy (simply referred as autophagy)[9,10].

In neurons, conditional knockout of Vps34 causes progressive synaptic loss followed by extensive gliosis and neurodegeneration[12,13]. While no human mutations in *PIK3C3*, the gene encoding Vps34, have been associated with neurodegenerative disorders, the relevance of the $PI3P/PI(3,5)P_2$ pathway in neurological disorders is supported by the existence of amyotrophic lateral sclerosis (ALS) and Charcot–Marie–Tooth 4 J (CMT4J) causing mutations in *FIG4* as well as PD causing mutations in *VAC14*, both of which are key components of the PIKfyve complex controlling $PI(3,5)P_2$ metabolism[11,14]. In addition, we have recently reported that PI3P is selectively deficient in the brain of patients with AD and mouse models thereof[15]. Furthermore, silencing Vps34 in primary neurons was shown to cause endosomal anomalies and altered amyloidogenic processing of amyloid precursor protein (APP), which are important pathological features of AD[1,2]. Overall, these studies implicate dysregulation of the PI3P pathway and associated endolysosomal perturbation in neurodegeneration.

In this study, we employ pharmacological inhibition and genetic ablation of Vps34 in mice to assess the impact of PI3P deficiency on neuronal autophagy and endolysosomal function, with emphasis on APP and lipid metabolism. We confirm that disruption of Vps34 function impairs endolysosomal function and report a profound alteration of cellular lipid metabolism consistent with a LSD. PI3P depletion promoted the physical disruption of endolysosomal membranes and secretion of exosomes harboring APP C-terminal fragments (APP-CTFs) and unique lipid signatures, including an enrichment of the endolysosomal phospholipid bis(monoacylglycero)phosphate (BMP) (also known as lysobisphosphatidic acid or LBPA)[16]. We show that release of these atypical exosomes is blocked by inhibition of neutral sphingomyelinase 2 (nSMase2) and de novo sphingolipid synthesis. Together, our results demonstrate that endolysosomal dysfunction triggers a homeostatic response leading to the secretion of atypical exosomes, allowing for elimination of lysosomal contents that cannot be efficiently degraded. They further highlight the potential of exosomes as biomarkers for neurodegenerative disorders involving endolysosomal dysfunction.

## Results

**PI3P depletion causes endosomal dysfunction**. To test the role of PI3P in endosomal function, we used the newly developed highly specific Vps34 kinase inhibitor VPS34IN1[17] (see also refs. [18,19]), which selectively decreased PI3P by ~50% after 24 h in the murine neuroblastoma line N2a (Supplementary Fig. 1a). Drug treatment increased Beclin 1 levels, but did not downregulate its other interacting partners from complex I, namely ATG14L and Vps15, in contrast to knockout of Vps34[20], ruling out indirect effects of destabilization of these proteins (Supplementary Fig. 1b). Importantly, a 24 h treatment with VPS34IN1 did not affect neuronal cell viability in vitro (Supplementary Fig. 1c).

Next, we investigated the impact of Vps34 inhibition on early endosomal trafficking in primary mouse cortical neurons. Pharmacological inhibition of Vps34 for 3 h caused a ~50% increase in the diameter of EEA1-positive endosomal puncta (Fig. 1a), in agreement with previous Vps34 silencing experiments in neurons[15]. EEA1 fluorescence intensity was decreased by ~30% after Vps34 inhibition (Fig. 1a), likely reflecting reduced membrane association of this PI3P-interacting protein[9,10]. This was confirmed in N2a cells fractionated in particulate and soluble fractions, showing higher levels of soluble EEA1, but not Rab5, in VPS34IN1-treated cells (Supplementary Fig. 1d). Rab5 puncta intensity and size were also significantly increased (Fig. 1a and Supplementary Fig. 1e). Upon Vps34 inhibition, Rab5 showed increased colocalization with APPL1 (Fig. 1b), a protein that associates with PI3P-negative early endosomes[21]. These observations confirm the key role of Vps34 kinase activity in early endosomal traffic in neurons.

**Vps34 inhibition slows lysosomal degradation of APP-CTFs**. Vps34 controls endolysosomal function in non-neuronal cells, in part by mediating sorting via the ESCRT pathway and delivery of hydrolases to lysosomes[10]. Here we found that neurons treated with VPS34IN1 for 4 h show delayed degradation of the EGF receptor following EGF stimulation (Supplementary Fig. 2a). The remaining degradation was blocked by V-ATPase inhibitor bafilomycin A1 (BafA1), suggesting that Vps34 inhibition only partially impairs lysosomal function (Supplementary Fig. 2b). Next, we assessed the maturation of Cathepsin D (CatD), a lysosomal hydrolase produced in the biosynthetic pathway as precursor proCatD (~50 kDa) and sorted to the endolysosomal compartment, where it is processed into a mature form (~30 kD) at acidic pH[22]. Vps34 inhibition for 24 h caused a defect in CatD maturation, based on the decreased CatD/proCatD ratio, but not in total levels (Fig. 1c). However, confocal analysis of CatD showed that its sorting into the endolysosomal lumen was not grossly affected (Fig. 1c).

We next investigated the impact of acute PI3P depletion on the processing of endogenous APP. The C-terminal fragments of APP α (APP-CTFα) and β (APP-CTFβ), which are produced by α-secretase and β-secretase, respectively, and further processed by γ-secretase, are known to accumulate as a result of lysosomal

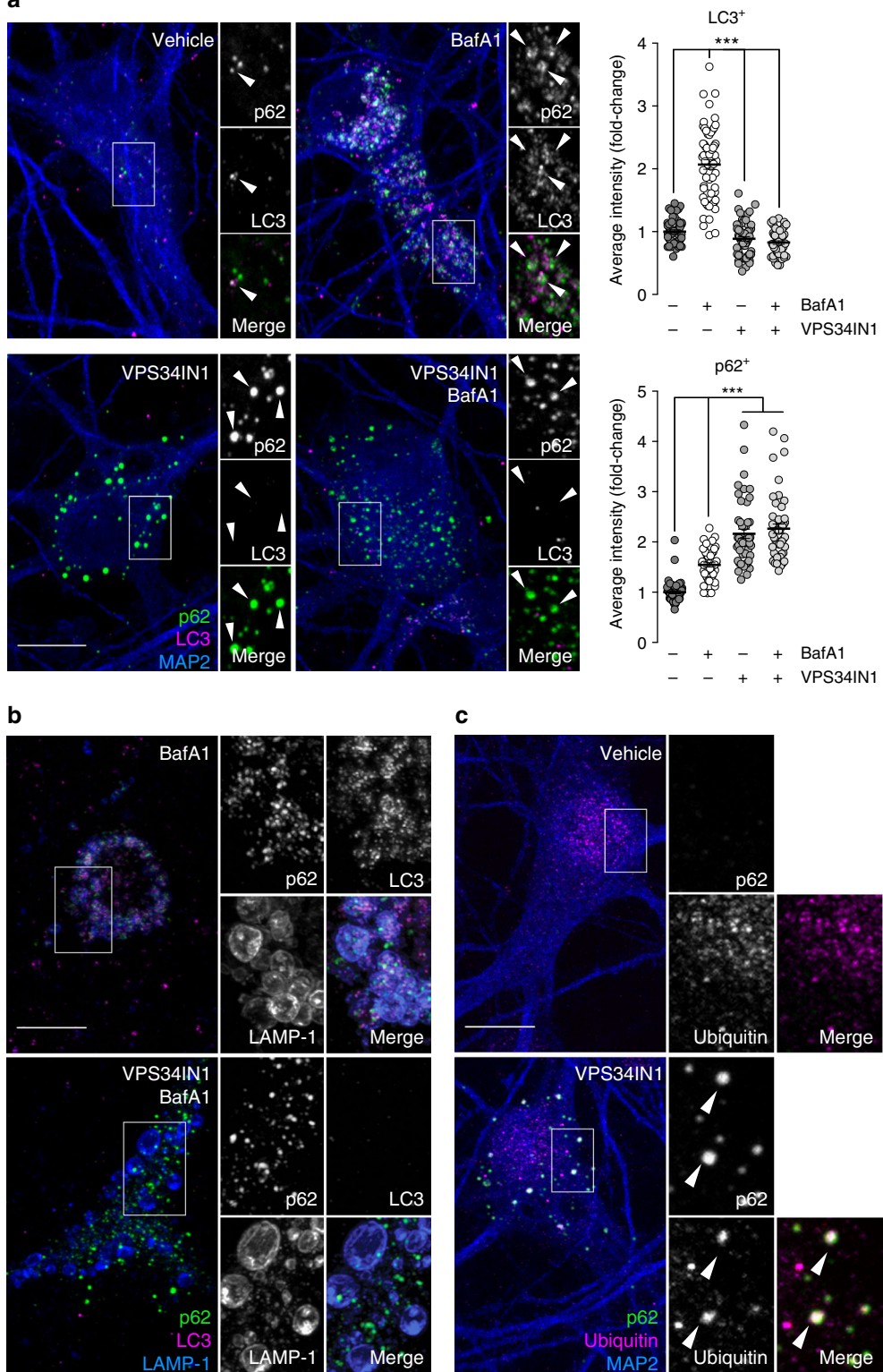

**Fig. 2** Vps34 inhibition blocks autophagy initiation and causes accumulation of ubiquitin-positive and p62-positive structures. **a** Representative confocal images of cortical neurons treated with vehicle, Bafilomycin A1 (BafA1) at 50 nM, VPS34IN1 at 3 μM or cotreated for 3 h. Arrows highlight LC3 and p62 structures. Right panel, bar graphs denote average object intensity, per cell (mean ± SEM, $N = 49$-60 cells, from three independent experiments). Scale bar, 10 μm. ***$p < 0.001$ in one-way ANOVA, Holm–Sidak's multiple comparisons test. **b** Representative confocal images of cortical neurons treated as in **a** and immunostained for LAMP-1, LC3, and p62. Airyscan insets highlight position of LC3 and p62 structures relative to LAMP-1-positive membranes. Scale bar, 10 μm. **c** Representative confocal images of cultured cortical neurons treated with vehicle or VPS34IN1 at 3 μM for 24 h. Arrows highlight p62 and ubiquitin colocalization. Scale bar, 10 μm

dysfunction[23]. Vps34 inhibition caused a ~40% increase in APP-CTFα/β (hereafter referred to as APP-CTFs) levels without altering levels of full-length APP (FL-APP; Fig. 1d). Of note, the BACE-1 cleavage product, APP-CTFβ, was also increased by ~40%. Contrary to effects observed upon Vps34 silencing[15], levels of secreted γ-secretase products Aβ40 and Aβ42 were decreased by ~20–25% after Vps34 inhibition (Fig. 1d). Blocking Vps34 in N2a cells caused a more robust increase in total APP-CTF levels and APP-CTFβ specifically, while Aβ40 and Aβ42 secretion was decreased by 40–65% (Fig. 1e).

To test if APP-CTF accumulation results from decreased lysosomal degradation rather than decreased γ-secretase

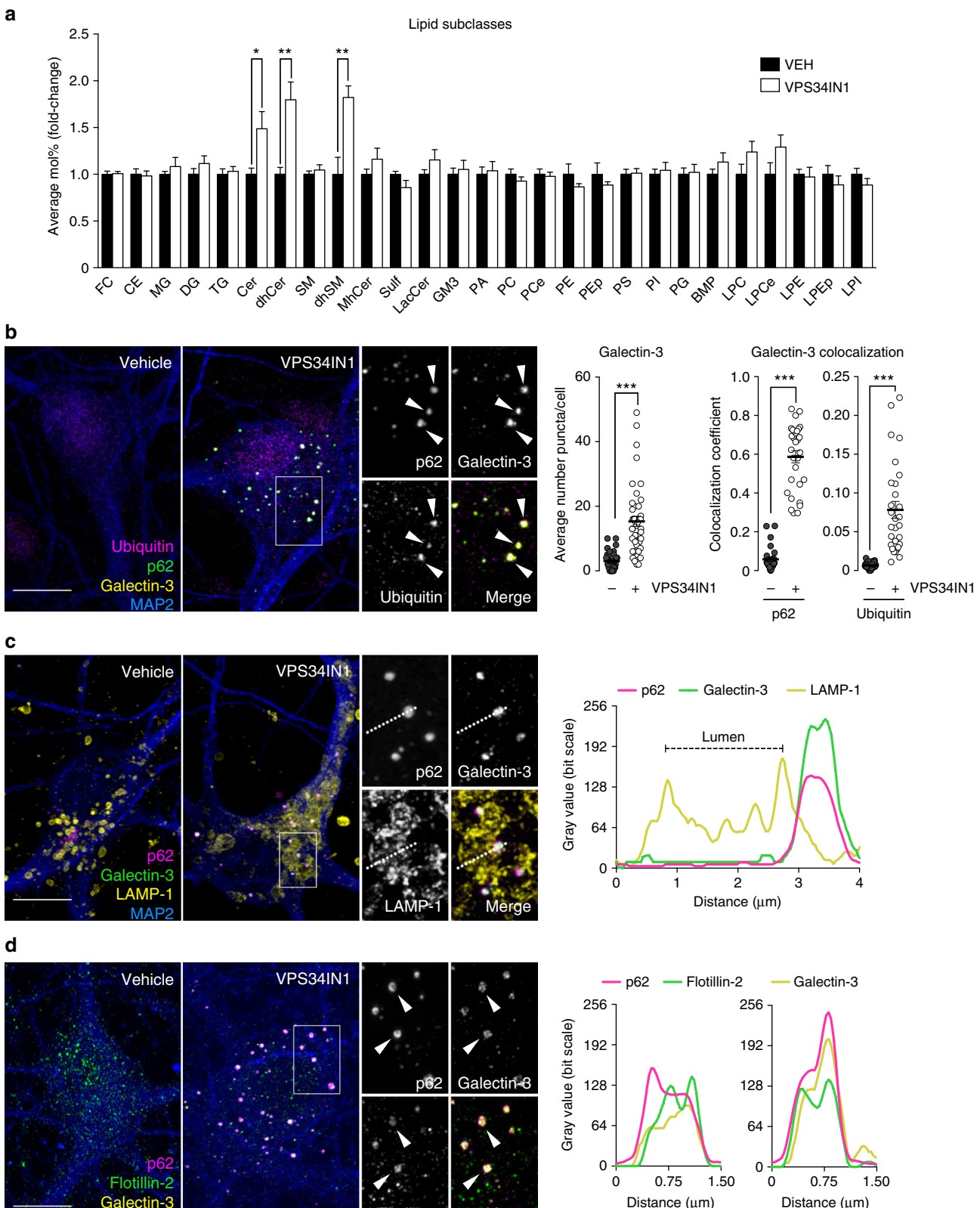

processing, we pulse-treated N2a cells with protein synthesis inhibitor cycloheximide and monitored FL-APP and APP-CTF levels for 6 h (Supplementary Fig. 2c). While VPS34IN1 did not affect FL-APP turnover, it caused a delay in APP-CTF degradation, particularly at 2 h (Supplementary Fig. 2c). To exclude processing defects by γ-secretase, cells were also co-incubated with the γ-secretase inhibitor, compound XXI (XXI γ-inhibitor) (Supplementary Fig. 2d). XXI γ-inhibitor extended APP-CTF half-life relative to cycloheximide alone (i.e., from 2 to 4 h), confirming that γ-secretase cleavage is a main pathway for the clearance of APP-CTFs. However, Vps34 inhibition still significantly delayed APP-CTF clearance despite γ-secretase inhibition (Supplementary Fig. 2d), confirming that a significant amount of APP-CTFs are degraded in lysosomes in a Vps34-dependent fashion.

**Vps34 inhibition blocks autophagy initiation.** Next, we assessed the autophagic flux by treating neurons for 3 h with BafA1 to block the clearance of autophagosomes in the presence or absence of VPS34IN1. While BafA1 alone caused a ~2-fold increase in the fluorescence of LC3-positive structures (Fig. 2a) and a ~5-fold increase in lipidated LC3 (LC3-II) by immunoblotting (Supplementary Fig. 3a), these effects were blocked by Vps34 inhibition, indicating inhibition of autophagosome formation. Next, we investigated the autophagy adapter p62, which delivers poly-ubiquitinated cargoes to growing autophagosomes via its ubiquitin-associated domain and LC3-interacting region. As an autophagy substrate, p62 accumulates and aggregates in ubiquitin-positive structures when autophagy is impaired[24]. Vps34 inhibition caused a ~2-fold increase in the fluorescence of p62-positive structures (Fig. 2a), similar to BafA1 alone or after combined treatments (Supplementary Fig. 3a). Remarkably, super-resolution Airyscan confocal images showed that p62-positive puncta do not accumulate in the lumen of LAMP-1-positive late endosomes/lysosomes upon Vps34 inhibition, while they were observed in those of BafA1-treated cells (Fig. 2b). This suggests that sorting of p62 structures into late endosomes/lysosomes requires PI3P-dependent autophagosome formation, a process that is not altered by BafA1. Electron microscopic (EM) analysis of VPS34IN1-treated cortical neurons revealed enlarged vacuoles with decreased intraluminal material and previously observed electron dense structures[25] in the vicinity of endocytic compartments (Supplementary Fig. 3b), which morphologically resemble LAMP-1 and p62-positive structures detected by immunofluorescence (Fig. 2b). Additionally, ubiquitin-positive inclusions co-localizing with p62 were revealed by confocal microscopy after a 24 h treatment with VPS34IN1 (Fig. 2c), consistent with the accumulation of poly-ubiquitinated proteins and p62 observed by immunoblotting (Supplementary Fig. 3c). These results were confirmed in primary cortical neurons lacking Vps34 as a result of lentiviral expression of *Cre* recombinase in a *Pik3c3*flox/flox background (Supplementary Fig. 3d, e) and therefore highlight the essential role of Vps34 and its kinase activity in neuronal autophagy.

**Lipid metabolism is severely affected by Vps34 inhibition.** Given the deleterious impact of Vps34 inhibition on endolysosomal/autophagic function, we hypothesized that a broader, secondary lipid dysregulation may result from PI3P deficiency. We used liquid chromatography–mass spectrometry (LC–MS) to analyze the lipid composition of cultured cortical neurons following Vps34 inhibition. Of all lipid classes, sphingolipids were the most significantly impaired, particularly ceramide (Cer), dihydroceramide (dhCer) and dihydrosphingomyelin (dhSM) (Fig. 3a), for which many molecular species were increased (Supplementary Fig. 4a). Additionally, levels of BMP were quantified. BMP is predominantly enriched in ILVs of late endosomes[16] and is known to be elevated in LSDs, such as NPC[26] as well as in the brain of AD patients[27]. Although no changes in total BMP levels were found after VPS34IN1 treatment, several molecular species of this phospholipid were increased (Fig. 3a; Supplementary Fig. 4b). While only diffuse immunostaining of BMP was detected in primary neurons, immunostaining of N2a cells confirmed the luminal localization of this phospholipid in LAMP-1 compartments in both control and treated conditions, suggesting that no gross alteration of BMP localization occurs in these organelles upon Vps34 inhibition, which also caused a clear accumulation of p62 in their vicinity (Supplementary Fig. 4c). Overall, these data indicate that reducing PI3P synthesis causes a secondary perturbation of lipid metabolism.

**Vps34 inhibition causes endolysosomal membrane damage.** Next, we tested if endolysosomal compartments are physically disrupted by Vps34 inhibition. We thus stained VPS34IN1-treated primary cortical neurons for galectin-3, a recently established marker for endolysosomal membrane damage[28], and observed an increase in the number of galectin-3-positive structures (Fig. 3b, c). Galectin-3 puncta greatly colocalized with p62, as well as ubiquitin, and were generally juxtaposed to and distinct from LAMP-1 compartments. In fact, super-resolution confocal and linescan analyses evidenced the segregation of p62/galectin-3 intensity peaks from LAMP-1 membranes and luminal exclusion of these structures (Fig. 3b, c). We also found colocalization between galectin-3, p62, and flotillin-2, a membrane-associated protein commonly used to identify cholesterol-enriched and sphingolipid-enriched microdomains[29] (Fig. 3d). Although APP-CTFs accumulate in VPS34IN1-treated neurons, immunostainings of APP and APP-CTFs did not colocalize with p62 (Supplementary Fig. 4d), suggesting that APP does not accumulate on damaged endolysosomal structures. Together with the finding that genetic ablation of Vps34 in cultured neurons also increases the number of galectin-3 puncta and their colocalization with p62 (Supplementary Fig. 5), our data suggest that Vps34 inhibition results in physical damage of endolysosomal membranes. Proximity to but lack of luminal sorting into LAMP-1 compartments indicates these damaged organelles are marked for degradation via ubiquitination and recruitment of autophagy adapter p62, but they are not efficiently eliminated by selective autophagy[28], likely from the reduced ability to lipidate LC3 upon Vps34 inhibition.

**Fig. 3** Vps34 inhibition causes cellular accumulation of sphingolipids and induces endolysosomal membrane damage. **a** LC–MS analysis of lipids extracted from primary cortical neurons treated with vehicle or VPS34IN1 at 3 μM for 24 h. For lipid nomenclature, see Methods section. Values are expressed as average Mol% of total lipid measured, normalized to vehicle (mean ± SEM, $N = 8$, from two independent experiments). *$p < 0.05$, **$p < 0.01$ in two-tailed Student's $t$ test. **b–d** Representative confocal images of cultured cortical neurons treated as in **a**. **b** Airyscan insets highlight triple colocalization between galectin-3, ubiquitin, and p62. Bar graphs indicate average number of galectin-3 puncta per cell (mean ± SEM, $N = 40$ cells, from three independent experiments), fraction of galectin-3 colocalizing with p62, and ubiquitin colocalizing with galectin-3 (mean ± SEM, $N = 30$ cells, from three independent experiments). **c** Airyscan insets highlight p62 and galectin-3 colocalization in close proximity to LAMP-1-positive membranes. Right panel, linescan intensity profile for adjacent p62/galectin-3 and LAMP-1 structures. **d** Airyscan insets and linescan intensity profile highlight triple colocalization between galectin-3, p62, and flotillin-2. Scale bar, 10 μm

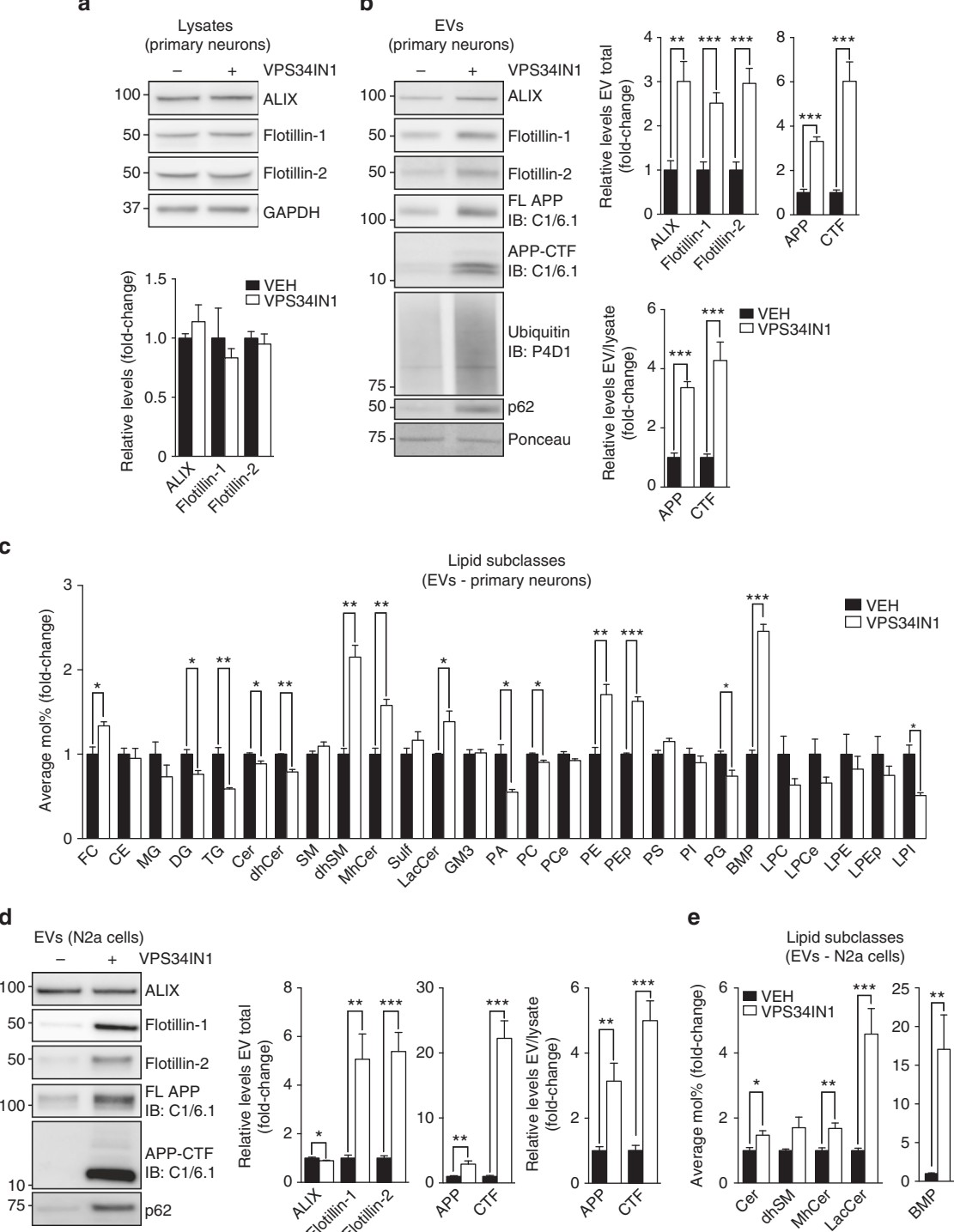

**Fig. 4** VPS34IN1 treatment causes secretion of extracellular vesicles (EVs) enriched for APP-CTFs, sphingolipids and BMP. **a**, **b** Western blot analysis of cell lysates and EVs from primary cortical neurons treated with vehicle or VPS34IN1 at 3 µM for 24 h. EV protein levels were normalized to lysate total protein (EV total) or lysate levels of the corresponding protein (EV/Lysate ratio). Bar graph denotes average protein levels normalized to vehicle (mean ± SEM, $N = 4$ for cell lysates, $N = 6$ for EVs, from two independent experiments). **$p < 0.01$, ***$p < 0.001$ in two-tailed Student's $t$ test. **c** LC–MS analysis of lipids extracted from EVs collected from primary cortical neuron culture media after treatment as in **a**. Values are expressed as average Mol% of total lipid measured, normalized to vehicle (mean ± SEM, $N = 3$, each from a pool of two biological replicates). *$p < 0.05$, **$p < 0.01$, ***$p < 0.01$ in two-tailed Student's $t$ test. **d**, **e** Western blot and LC–MS analysis of lipids extracted from EVs collected from N2a cell culture media after treatment with vehicle or VPS34IN1 at 1 µM for 24 h. EV protein levels were normalized to lysate total protein (EV total) or lysate levels of the corresponding protein (EV/Lysate ratio). For complete lipid panel, see Supplementary Fig. 6d (mean ± SEM, $N = 6$, from two independent experiments). Lipid values are expressed as average Mol% of total lipids measured, normalized to vehicle (mean ± SEM $N = 7$, each from a pool of two biological replicates) *$p < 0.05$, **$p < 0.01$, ***$p < 0.001$ in two-tailed Student's $t$ test

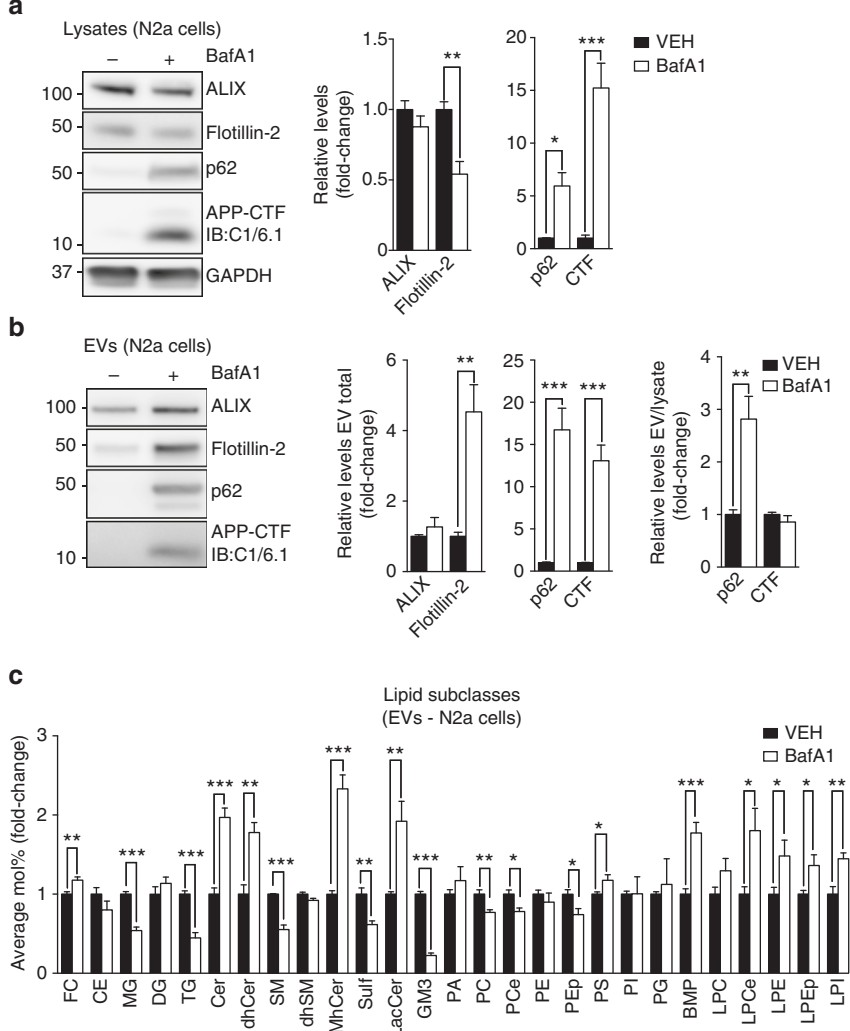

**Fig. 5** Inhibition of V-ATPase promotes secretion of EVs enriched for APP-CTF and BMP. **a**, **b** Western blot analysis of cell lysates and EVs from N2a cells treated with vehicle or BafA1 at 25 nM for 24 h. EV protein levels were normalized to lysate total protein (EV total) or lysate levels of the corresponding protein (EV/Lysate ratio). Bar graph denotes average protein levels normalized to vehicle (mean ± SEM, $N = 4$ for lysates, $N = 6$ for for EVs, from two independent experiments). **$p < 0.01$, ***$p < 0.001$ in two-tailed Student's $t$ test. **c** LC–MS analysis of lipids extracted from EVs collected from N2a cells treated as in **a**. Values are expressed as average Mol% of total lipids measured, normalized to vehicle (mean ± SEM, $N = 5$, each from a pool of two biological replicates) *$p < 0.05$, **$p < 0.01$, ***$p < 0.001$ in two-tailed Student's $t$ test

Whether these damaged organelles are early endosome or late endosomes/lysosomes that have lost their membrane markers through proteolysis as a result of membrane damage remains unclear.

**APP-CTFs are secreted in BMP-enriched exosomes**. Lysosomal stress has been hypothesized to cause the release of extracellular vesicles (EVs), such as exosomes, as an alternative pathway mediating the elimination of cellular waste, including toxic protein aggregates and lipids[30–32]. Exosomes are small 40–100 nm vesicles that are released when multivesicular endosomes fuse with the plasma membrane[33].

The association of flotillin-2 with damaged endolysosomal membranes and the fact that flotillins have been reported on EVs[33] prompted us to determine if VPS34IN1-induced endolysosomal dysfunction promotes EV release. EVs were purified by filtration and ultracentrifugation of cell media[34,35] (Supplementary Fig. 6a, b). In primary cortical neurons, Vps34 inhibition increased secretion of three EV markers (ALIX, Flotillin-1, and

Flotillin-2) by ~3-fold, while levels of the same markers were unaffected in cell lysates (Fig. 4a, b). Given the effect of Vps34 blockade on APP metabolism, we examined levels of FL-APP and APP-CTFs in EVs (Fig. 4b). Remarkably, APP-CTFs were increased to a greater extent than FL-APP (6-fold vs. 3-fold, respectively), suggesting that APP-CTFs are sorted more efficiently to EVs. However, since APP-CTFs levels were also higher in lysates from VPS34IN1-treated cells (Fig. 1d), the fold increase in EV secretion was comparable for FL-APP and APP-CTFs when normalized to lysate levels of the same proteins (Fig. 4b). Since poly-ubiquitinated proteins as well as p62 have been previously reported on EVs[36,37], we blotted the EV fractions with anti-ubiquitin and anti-p62 antibodies and found increased levels upon VPS34IN1 treatment, supporting the hypothesis that EVs can be used as a vehicle for the disposal of undigested material (Fig. 4b).

We next analyzed EV lipid composition by LC–MS. EVs from VPS34IN1-treated cortical neurons were enriched for cholesterol and specific sphingolipid subclasses, namely dhSM, monohexosylceramide (MhCer) and lactosylceramide (LacCer) (Fig. 4c)

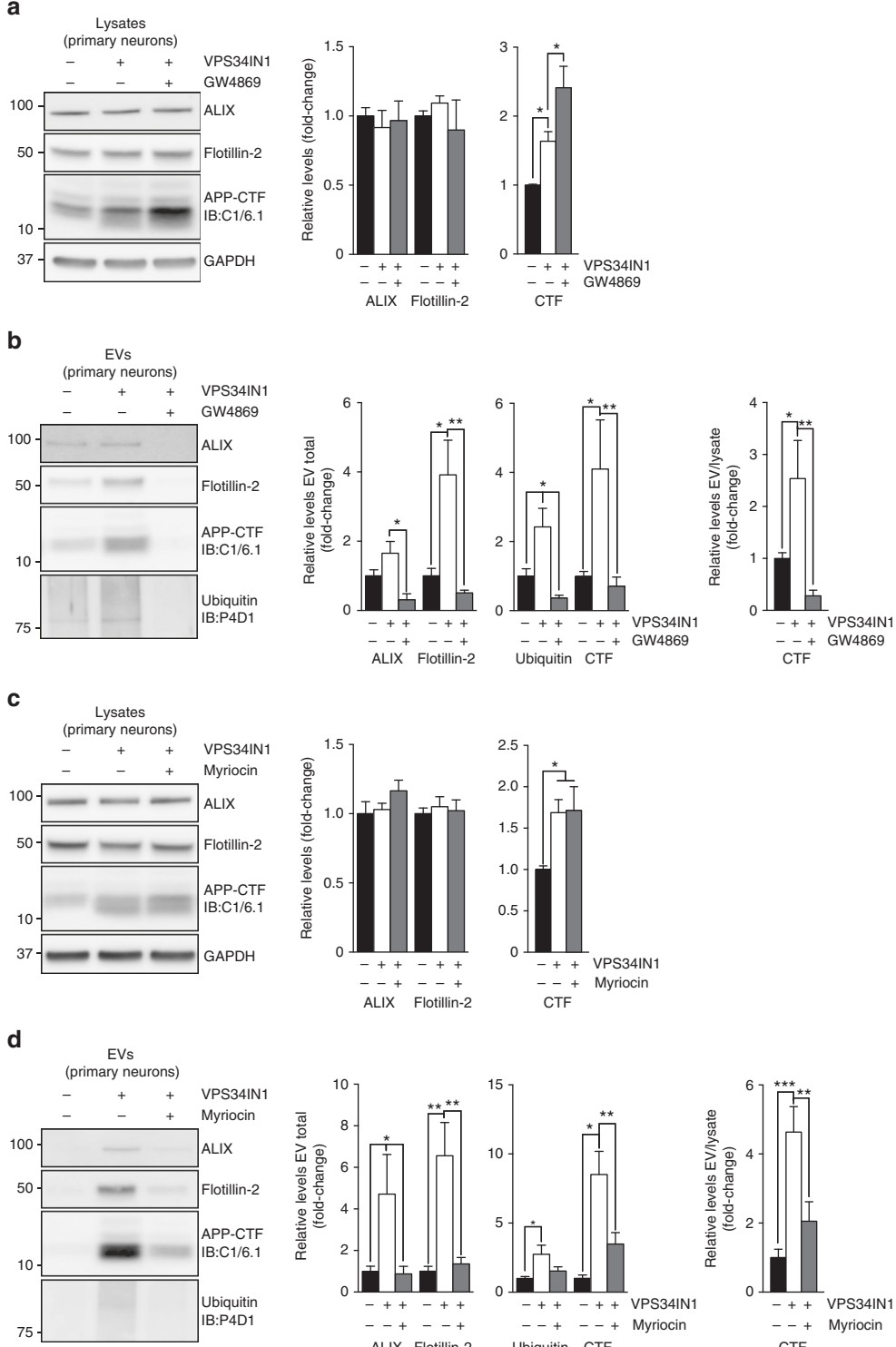

**Fig. 6** Pharmacological blockade of neutral Sphingomyelinase 2 or inhibition of de novo sphingolipid synthesis inhibit VPS34IN1-induced release of exosomal APP-CTFs. **a–d** Western blot analysis of cell lysates and EVs from cortical neurons treated with vehicle, VPS34IN1 at 3 μM or cotreated with GW4869 at 10 μM or Myriocin at 1 μM for 36 h. EV protein levels were normalized to lysate total protein (EV total) or lysate levels of the corresponding protein (EV/lysate ratio). Bar graph denotes average protein levels normalized to vehicle (mean ± SEM, $N = 5$ for vehicle, $N = 6$ for treated groups, from two independent experiments). *$p < 0.05$, **$p < 0.01$, ***$p < 0.001$ in one-way ANOVA, Holm–Sidak's multiple comparisons test

relative to controls. The most robust change, however, was a ~2.5-fold increase in total BMP (Fig. 4c), reflecting increase in multiple molecular species of this phospholipid (Supplementary Fig. 6c). These results were validated in N2a cells, as EV-associated APP-CTFs, FL-APP, flotillin-1, and -2 levels were all increased (Fig. 4d, e). Of note, proteins typically associated with the number of exosomes, ALIX and CD63, or ESCRT-dependent ILV sorting, Tsg101 and Hrs, were minimally affected, suggesting that Vps34 inhibition likely affects composition rather than quantity of EVs in an ESCRT-independent fashion (Fig. 4d and

data not shown). Lipidomic profiling showed an increase in an overlapping set of sphingolipid subclasses, such as MhCer and LacCer, and a striking ~20-fold increase in BMP levels, including all individual molecular species detected (Fig. 4e, Supplementary Fig. 6d). Additionally, while a ~70% increase in dhSM levels was also detected ($p = 0.05$), Cer levels were increased by ~50% in

contrast to the ~10% decrease found in primary neuron-derived EVs (Fig. 4c). These results were also confirmed using a chemically-distinct Vps34 kinase inhibitor, SAR405[19], indicating on-target effects (data not shown). Altogether, our observations suggest that EVs secreted upon PI3P depletion result from endolysosomal/autophagic dysfunction and are exosomes, based

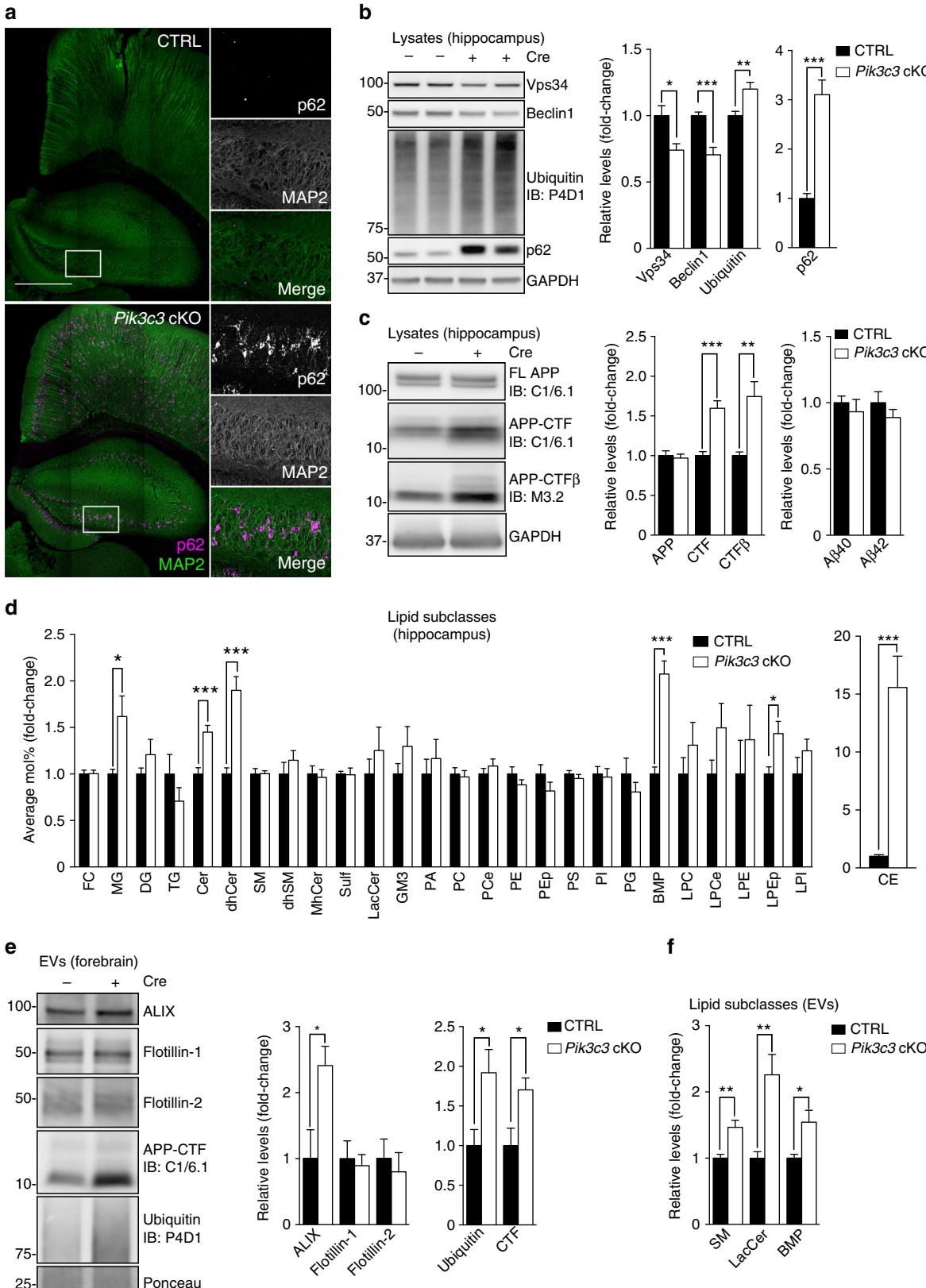

on the fact that BMP is associated with ILVs[16] and is thus a bona fide marker for these organelles[16].

**APP-CTF secretion requires endolysosomal dysfunction**. To investigate if APP-CTF exosomal secretion merely reflects intracellular accumulation, we compared the effect of Vps34 inhibition to γ-secretase inhibition, which dramatically increases cellular APP-CTF levels, but does not cause major alterations of the endolysosomal system[38]. N2a cells were treated for 24 h with the XXI γ-inhibitor and VPS34IN1, either singly or in combination. Cellular APP-CTF levels were increased by XXI γ-inhibitor to a higher extent than VPS34IN1 alone, while they were additively increased by combined treatment (Supplementary Fig. 7a). Consistent with a lack of impact on autophagic/lysosomal function, the XXI γ-inhibitor alone did not affect total p62 levels (Supplementary Fig. 7a). Remarkably, despite the robust accumulation of APP-CTFs observed after XXI γ-inhibitor[39], a smaller proportion was released on EVs relative to vehicle or VPS34IN1 treatment (Supplementary Fig. 7b, low and high exposure panels). Combined treatments caused an additive increase in EV-associated APP-CTFs relative to VPS34 inhibition alone, but to a lesser extent than that observed in lysates (Supplementary Fig. 7b). These data suggest that active sorting of APP-CTFs into EVs occurs as a result of Vps34 inhibition and that the amount of EV-associated APP-CTF does not simply reflect cellular APP-CTF levels.

Next, we sought to precisely delineate the role of autophagy blockade vs. endolysosomal dysfunction by VPS34IN1 on exosomal secretion of APP-CTFs. To specifically assess the contribution of autophagy, we used CRISPR-Cas9 gene-edited N2a cells lacking *Atg5* (*Atg5* KO), a core component of the autophagy machinery[40]. *Atg5* ablation was confirmed by the absence of Atg12-Atg5 conjugates (Supplementary Fig. 7c). *Atg5* KO N2a cell lysates showed increased basal cellular levels of p62 and APP-CTFs relatively to isogenic controls. Interestingly, Vps34 inhibition caused a similar fold increase in cellular p62 and APP-CTF levels in both cell lines, in addition to baseline effects induced by *Atg5* KO (Supplementary Fig. 7c). While *Atg5* KO alone showed a trend for increased secretion of the EV markers analyzed, VPS34IN1 treatment caused a much more dramatic effect in EV-associated APP-CTFs and a comparable fold increase in both naive and *Atg5* KO cells (Supplementary Fig. 7d). Therefore, we conclude that the increase of APP-CTF secretion via EVs induced by Vps34 inhibition occurs independently of major autophagy defects and suggests it originates from other aspects of endolysosomal dysfunction.

We also investigated whether exosomal APP-CTF levels are affected by endolysosomal alkalization induced by treating N2a cells with BafA1 for 24 h. Western blot analysis of cell lysates showed that BafA1 treatment causes a dramatic increase in p62 levels as well as in APP-CTFs (Fig. 5a). In contrast, cellular flotillin-2 levels were decreased by ~50% upon BafA1 treatment. BafA1-treated cells showed a striking increase in flotillin-2, p62, and APP-CTFs in EVs, although the latter was proportional to cellular levels (Fig. 5b). In addition, LC–MS analysis of EVs from BafA1-treated cells showed a ~20% enrichment for cholesterol and a ~2-fold increase in the sphingolipid subclasses Cer, dhCer, MhCer, and LacCer (Fig. 5c). BMP levels were increased by ~2-fold as seen in N2a cells and primary neurons treated with VPS34IN1 (Fig. 5c). Altogether, these results suggest that release of EVs enriched for APP-CTFs and lipids of late endocytic/lysosomal compartments is intrinsically related to endolysosomal dysfunction.

**Exosome secretion is modulated by ceramide synthesis**. Cer production by nSMase2 has been directly implicated in exosome biogenesis and secretion[41,42]. Considering the impact of Vps34 inhibition on Cer metabolism, we tested if GW4869, a nSMase2 inhibitor, affects exosomal secretion of APP-CTFs induced by VPS34IN1. Cotreatment of primary cortical neurons with VPS34IN1 and GW4869 increased the cellular accumulation of APP-CTFs relative to VPS34IN1 alone (Fig. 6a). In contrast, no changes were observed for cellular flotillin-2 levels (Fig. 6a). GW4869 caused an overall decrease in EV secretion, based on the reduction of ALIX, flotillin-2, APP-CTFs and poly-ubiquitinated protein levels observed in EV fractions (Fig. 6b). The effect of nSMase2 inhibition on exosomes was largely phenocopied by treatment with myriocin, which inhibits serine palmitoyl transferase, i.e., the enzyme catalyzing the first step in de novo sphingolipid synthesis (Fig. 6c, d). These results suggest that exosome secretion quantitatively alleviates intracellular burden and implicate sphingolipid metabolism in the sorting and secretion of APP-CTFs in exosomes.

**Vps34 ablation in neurons alters brain exosome composition**. To validate the role of neuronal Vps34 in endolysosomal function and exosome secretion in vivo, we conditionally deleted the gene encoding Vps34 in forebrain excitatory pyramidal neurons by crossing *CaMKII-Cre* transgenic mice with *Pik3c3*^flox/flox mice (*Pik3c3* cKO). Mutant mice show extensive gliosis and progressive neuronal loss in the hippocampus and cortex, as described previously[12]. Here, mouse hippocampi were analyzed at 2 months of age, approximately a month after *Cre* expression in the forebrain and prior to neurodegeneration, as shown with comparable MAP2 stainings in neurons from both genotypes, in contrast with 3-month-old mice which show a profound decrease in MAP2 stainings in *Pik3c3* cKO brain (Fig. 7a and Supplementary Fig. 8a)[12]. Total Vps34 and Beclin 1 protein levels were decreased by ~30% in *Pik3c3* cKO mice, with the remaining expression likely resulting from glial cells as well as inhibitory neurons (Fig. 7b). As expected, *Pik3c3* cKO mice showed accumulation of poly-ubiquitinated proteins and p62, indicating endolysosomal/autophagic defects (Fig. 7b). As seen in vitro, Vps34 deficiency increased levels of APP-CTFs (including APP-CTFβ) but not FL-APP (Fig. 7c). However, hippocampal Aβ40 and Aβ42 levels were unchanged (Fig. 7c).

**Fig. 7** Neuronal Vps34 deficiency in vivo leads to endolysosomal dysfunction and increased exosomal APP-CTF sorting. **a** Brain sections from 2-month-old *Pik3c3*^flox/flox (CTRL) and *Pik3c3*^flox/flox; *CaMKII-Cre* (*Pik3c3* cKO) mice immunostained for MAP2 and p62. Scale bar, 500 μm. **b, c** Western blot analysis of hippocampus lysates from 2-month-old CTRL and *Pik3c3* cKO mice (mean ± SEM, *N* = 7 and 8, respectively in **b**; *N* = 8 in **c**, from two independent experiments). **c** Right panel, soluble Aβ40 and Aβ42 levels in hippocampus lysates of 2-month-old CTRL and *Pik3c3* cKO mice (mean ± SEM, *N* = 9 mice, from two independent experiments). **p* < 0.05, ***p* < 0.01, ****p* < 0.001 in two-tailed Student's *t* test. **d** LC–MS analysis of lipids extracted from hippocampus lysates of 2-month-old CTRL and *Pik3c3* cKO mice. Values are expressed as average Mol% of total lipids measured, normalized to CTRL. (mean ± SEM, *N* = 8 and 9, respectively, from two independent experiments). **p* < 0.05, ****p* < 0.001 in two-tailed Student's *t* test. **e** Western blot analysis of EVs isolated from the forebrain of 2-month-old CTRL and *Pik3c3* cKO mice. Bar graph denotes average protein levels normalized to CTRL (mean ± SEM, *N* = 4 mice, from two independent experiments). **p* < 0.05 in two-tailed Student's *t* test. **f** LC–MS analysis of EVs from 2-month-old CTRL and *Pik3c3* cKO mice. Values are express ed as average Mol% of total lipids measured, normalized to CTRL. For complete lipid panel and BMP species analysis, see Supplementary Fig. 8e. (mean ± SEM, *N* = 6 mice, from two independent experiments). **p* < 0.05, ***p* < 0.01 in two-tailed Student's *t* test

Next, we characterized the hippocampal lipidome. As observed in vitro, sphingolipid metabolism was affected with increased Cer and dhCer in *Pik3c3* cKO relative to control mice (CTRL) (Fig. 7d). BMP levels were also increased, consistent with endolysosomal impairment. We also found an increase in monoacylglycerol (MG) and, more striking, in cholesterol esters (CE), perhaps reflecting reactive gliosis and phagocytic activity[43].

To investigate the secretion of APP-CTFs via EVs, brain exosomes were purified by step gradient fractionation[42,44] (Supplementary Fig. 8b, c, d). Analysis of exosomal markers revealed a ~2.5-fold increase in ALIX but no changes in flotillin-1 or flotillin-2 in exosomes from *Pik3c3* cKO brain, relative to controls (Fig. 7e). Importantly, we observed a ~2-fold increase in exosomal poly-ubiquitinated proteins and APP-CTFs in *Pik3c3* cKO brain (Fig. 7e). FL-APP and p62 were not detected in purified exosomes, in contrast to those purified in vitro. Furthermore, lipidomic analysis of brain EVs revealed an increase in total SM, LacCer and BMP, with an increase in most BMP species detected (Fig. 7f and Supplementary Fig. 8e). While changes were generally more subtle than those observed in vitro, a more global alteration of phospholipids was found in hippocampal exosomes as well as increases in CE and MG, as seen in hippocampal tissue (Fig. 7d). Altogether, these results confirm that ablation of Vps34 in neurons in vivo induces endolysosomal dysfunction and lipid dysregulation, which are associated with the release of exosomes enriched for APP-CTFs and various lipids, including BMP.

## Discussion

Based on our previous work showing a deficiency of PI3P in AD brain[15], we have investigated the impact of neuronal Vps34 kinase inhibition and genetic ablation on endolysosomal function, autophagy, and APP metabolism. We found that blocking Vps34 leads to a profound dysregulation of lipid metabolism and endolysosomal membrane disruption, accumulation, and secretion of APP-CTFs on a subpopulation of exosomes also enriched for ubiquitinated proteins and atypical lipids such as BMP, demonstrating that neurons have the capacity to eliminate undigested, potentially toxic endolysosomal cargoes via exosomes.

Dysregulation of the endocytic pathway in neurons is increasingly viewed as one of the earliest pathological events in the pre-clinical stage of LOAD, even preceding autophagy disturbances also reported in this disease[45]. In this study, we show that Vps34 inhibition recapitulates some of the endolysosomal defects seen in early stages of AD, namely enlargement of EEA1-positive early endosomes[46], stabilization of APPL1/Rab5-positive endosomes and elevated APP-CTFβ[47]. Interestingly, Vps34 kinase inhibition led to a decrease in Aβ secretion in contrast to the previously reported increase in secreted Aβ resulting from partial knockdown of Vps34 in neurons[15]. This discrepancy may reflect kinase-dependent effects vs. effects related to the destabilization of the Vps34 complex, including Beclin 1[20], resulting from loss of Vps34 protein scaffold. In fact, our results are supported by previous findings reporting that low specificity PI3K inhibitors decrease Aβ secretion[48]. In addition, silencing ESCRT component Hrs, which mediates PI3P-dependent sorting of ubiquitinated cargoes into ILVs, drastically impairs Aβ secretion in N2a cells[49]. However, the most striking APP-related phenotype observed in cultured neurons as well as in vivo is the accumulation of APP-CTFs, which is reminiscent of NPC[50].

A key finding reported in this study is that reduction of PI3P causes lipid dysregulation and physically damaged endolysosomal membranes, as demonstrated by the enrichment of galectin-3 and flotillin-2 on p62-positive and ubiquitin-positive structures. Because these structures were also detected in Vps34 KO primary

neurons (which present lower Beclin 1 levels[20]), we exclude that secondary effects of pharmacological inhibition of Vps34, namely upregulation of Beclin 1, underlie these phenotypes. While the exact mechanism affecting endolysosomal integrity is unknown, it likely results from lysosomal substrate accumulation. The aberrant accumulation of sphingolipids, such as dihydrosphingolipids and Cer[51,52], and sterols[53] can potentially decrease lysosomal enzymatic activity and destabilize endolysosomal membranes. Alternatively, deficient trafficking or glycosylation of lysosomal membrane proteins may decrease protection from hydrolases and compromise membrane integrity[54]. This phenotype may be striking after Vps34 inhibition since both autophagy and sorting of p62 into lysosomes are reduced, thus preventing clearance of these damaged organelles by lysophagy-like mechanisms[28]. Relevant for neurodegenerative disorders, extracellular aggregates of proteins such as α-synuclein[55] and tau[56,57] have been shown to cause endolysosomal membrane rupture upon internalization, perhaps facilitating their cell-to-cell transmission. One of the potential implications of our study is that endolysosomal membrane destabilization caused by PI3P deficiency may sensitize neurons to aggregate-induced toxicity and pathology spreading.

Another important finding is the striking secretion of APP-CTFs in exosomes derived from neurons undergoing endolysosomal stress. Exosomes are increasingly associated with the transmission of aggregation-prone proteins[58], but little is known about the molecular mechanisms regulating their biogenesis, cargo selection and secretion[33]. Recent studies have shown that lysosomal alkalization by chloroquine promotes secretion of exosomes harboring α-synuclein[59] as well as the intracellular domain of APP, but not APP-CTFs[60]. While the latter result appears to be at odds with our BafA1 findings, it can be potentially explained by the fact that BafA1 causes an irreversible increase in lysosomal pH, unlike chloroquine. In fact, we found that neither chloroquine nor ammonium chloride increase exosomal levels of APP-CTFs, despite causing increases in their cellular levels (data not shown). Importantly, the fact that BafA1 partially phenocopies VPS34IN1 treatment suggests that blocking Vps34 may affect the V-type ATPase, which will be tested in future work. We also found that exosomal APP-CTF secretion upon Vps34 inhibition largely occurs independently of the core autophagy machinery, as Atg5-deficient N2a cells accumulate APP-CTFs intracellularly but do not secrete them to the same extent as Vps34-inhibited cells. Similarly, γ-secretase inhibition also fails to secrete large amounts of APP-CTFs on exosomes, despite their cellular accumulation, providing additional evidence that endolysosomal dysfunction, rather than intracellular APP-CTF accumulation per se, is the mechanism driving exosomal secretion of APP-CTFs. This process may minimize intracellular processing of these APP metabolites by γ-secretase, thus reducing Aβ generation and toxicity.

Our previous work showed that APP and APP-CTFs are sorted into the ILVs of multivesicular endosomes through ESCRT machinery in a pathway requiring PI3P and APP ubiquitination[10,15]. Of note, a subset of ILVs are known to be PI3P-positive within these endosomes[16]. Because APP-CTFs are coenriched with BMP on exosomes induced by Vps34 inhibition, PI3P deficiency may shunt APP-CTFs into a distinct subpopulation of ILVs which are BMP-positive and committed to the exosomal pathway. This is consistent with the longstanding view that PI3P-positive and BMP-positive ILVs are distinct within multivesicular endosomes[16]. Importantly, it agrees with the view that PI3P/ESCRT-independent pathways contribute to ILV biogenesis, including the nSMase2 pathway[41] as well as BMP itself[16]. Supporting this view, APP-CTFs lacking ubiquitination sites in the cytodomain[61], thus preventing sorting via the ESCRT pathway[15], are released through exosomes as efficiently as wild-type APP-

CTF upon Vps34 inhibition (data not shown). Interestingly, a recent study showed that inhibition of autophagosome–lysosomal fusion by tetraspanin-6 (TSPAN6) overexpression slowed the degradation of APP-CTFs and increased secretion of EVs in a process requiring syntenin[62]. This protein has been shown to interact with ALIX, a modulator of BMP biological functions, which include regulation of lysosomal lipases and storage of endolysosomal cholesterol[16]. Whether TSPAN6-dependent exosomes share the same biological properties as those induced by Vps34 inhibition is unclear. We also note that other studies have not reported a BMP enrichment in exosomes[63,64], likely reflecting the fact that BMP-enriched exosomes are not secreted constitutively, but rather released under endolysosomal stress.

Our study has also critical implications for biomarker discovery. Indeed, increased BMP levels are a common feature of several LSDs, including NPC[26] and have been reported in the brain of patients with AD[27] and Lewy body disease (LBD)[65], both of which associated with endolysosomal dysfunction[1]. Our study suggests that detection of BMP in bodily fluids, such as blood, cerebrospinal fluid (CSF), and urine, may be more indicative of exosomal secretion in response to LSD than a simple measure of phospholipidosis[66]. APP-CTFs have also been reported in human CSF[67], suggesting that they can be helpful biomarkers in clinical settings. In sum, APP-CTFs and BMP should be explored as exosome-related biomarkers for a range of neurodegenerative disorders, including AD, PD, LBD, FTD, ALS, and many LSDs.

Finally, an important question is whether exosomes released upon endolysosomal dysfunction are endowed with specific cell non-autonomous functions. Changes in EVs collected from *Pik3c3* cKO brain were subtler than those found in vitro, likely reflecting the dynamic turnover and metabolism of exosomes in vivo. For instance, they could convey "eat-me" signals directed to microglia, ensuring the proper elimination of unhealthy neurons through efferocytosis[68] as well as of the neuronal waste they carry. They could also present danger-associated molecular patterns ("DAMPs") affecting immune receptors[68]. Alternatively, they could be vectors for the aggregation and/or spreading of pathological proteins, including aberrant tau and Aβ[42,69], or constitute intracellular antigens triggering autoimmune responses[69].

In summary, our study reveals a specific homeostatic response counteracting endolysosomal dysfunction via secretion of atypical exosomes to eliminate lysosomal waste and define exosomal APP-CTFs and BMP as candidate biomarkers diagnostic of endolysosomal dysfunction associated with neurodegenerative disorders.

## Methods

**Reagents and antibodies.** The following compounds were used: BafA1 (25 or 50 nM, 023–11641, Wako), cycloheximide (50 μg/ml, C4859, Sigma-Aldrich), EGF (200 ng/ml, 01–101, EMD Millipore), GW4869 (10 μM, 13127, Cayman Chemicals), SAR405 (1 μM, HY-12481, MedChem Express), VPS34IN1 (1 or 3 μM, Dundee University), γ-secretase Inhibitor XXI, compound E (2 μM, 565790, EMD Millipore). All compounds were dissolved in dimethyl sulfoxide (DMSO) and control cells treated with 0.01% DMSO (Sigma-Aldrich). The following antibodies were purchased from commercial sources: ALIX (pab0204, Rabbit (Rb), 1:2000 in Western Blot (WB) of cell lysates and EVs, Covalab), ALIX (ABC40, Rb, 1:1000 in WB of forebrain EVs, EMD Millipore), amyloid precursor protein C1/6.1 C-terminal fragment (802801, Mouse (Ms), Biolegend, 1:500 in WB), APPL1 (ab59592, Rb, Abcam, 1:200 in IF and 1:1000 in WB), Atg12-Atg5 (2011, Rb, Cell Signaling, 1:1000 in WB), Beclin 1 (3738, Rb, Cell Signaling, 1:1000 in WB), β-Amyloid M3.2 (805701, Ms, Biolegend, 1:500 in WB), BIII-Tubulin (801201, Ms, Biolegend, 1:1000 in immunofluorescence (IF)), Calnexin (ab31290, Ms, Abcam, 1:200 in WB), EEA1 (sc-6415, Goat, Santa Cruz, 1:300 in IF), EEA1 (C45B10, Rb, Cell Signaling, 1:1000 in WB), EGFR (06-847, Rabbit (Rb), EMD Millipore, 1:1000 in WB), Flotillin-1 (610820, Ms, BD Biosciences, 1:1000 in WB), Flotillin-2 (610383, Ms, BD Biosciences, 1:50 for IF and 1:1000 in WB), GAPDH (MCA-1D4, Ms, Encor Biotech, 1:4000 in WB), Galectin-3 (sc-23938, Rat, Santa Cruz, 1:50 in IF), GM130 (610823, Ms, BD Biosciences, 1:500 in WB), LAMP-1 (1D4B, Rat, Developmental Studies Hybridoma Bank, 1:400 in IF), LAMP-1 (ab24170, Rb, Abcam, 1:400 in IF and 1:500 in WB), LC3 (M152-3, Ms, MBL, 1:300 in IF), LC3

(NB100-2220, Rb, Novus Biomedical, 1:1000 in WB), MAP2 (ab5392, Chicken, Abcam, 1:2000 in IF), p62 (03-GP62-C, Guinea Pig, American Research Productions, 1:1000 for IF and WB), Rab5 (108011, Ms, Synaptic Systems, 1:100 in IF and 1:500 in WB), Rab7 (9367, Rb, Cell Signaling, 1:1000 in WB), Ubiquitin (sc-8017, Ms, Santa Cruz, 1:1000 in WB, 1:100 in IF), Tubulin (T6074, Ms, Sigma-Aldrich, 1:5000 in WB), Vps15 (H00030849-M02, Ms, Abnova, 1:1000 in WB), Vps34 (4263, Rb, Cell Signaling, 1:1000 in WB). Antibodies raised against ATG14L (Rb, 1:500 in WB) were a generous gift from Dr. Zhenyu Yue (Icahn School of Medicine at Mount Sinai), anti-Cathepsin D (Rb, 1:10.000 in WB, 1:200 in IF) were a generous gift from Dr. Ralph Nixon (Nathan Kline Institute) and anti-BMP/LBPA (Ms, 1:50 in IF) were a generous gift from Dr. Jean Gruenberg (University of Geneva).

**Cell culture.** Murine neuroblastoma N2a cells were maintained at 37 °C in a humidified 5% $CO_2$ atmosphere in DMEM with GlutaMAX supplemented with 10% fetal bovine serum and penicillin (100 U/ml), streptomycin (100 μg/ml) (Thermofisher). Cells were negative for mycoplasma contamination. CRISPR-Cas9 gene-edited ATG5 KO and isogenic N2a cells were a kind gift from Dr. Hermann Schaetzl (University of Calgary). 24 h before drug treatment, cells were plated at 50% confluence. Primary cortical neurons were generated from newborn wild-type C57BL/6 or *Pik3c3*flox/flox mice. Briefly, cortices were dissected and chemically digested in 0.25% trypsin for 20 min at 37 °C. Cells were dissociated with a Pasteur pipette, plated on poly-ornithine-coated dishes or glass coverslips at a density of 50,000–100,000 cells/cm$^2$ and allowed to mature in Neurobasal-A supplemented with 2 mM Glutamax and 2% B27 (Thermofisher). Vps34 KO neurons were cultured from *Pik3c3*flox/flox mice and infected with lentivirus carrying catalytically active *Cre* recombinase or catalytically dead *Cre* (ΔCre)[20] after 7 days in vitro and grown up to 15 days. Lentiviruses were generated by transfecting HEK-293T with lentiviral vectors and pPACK-H1 packaging mix (System Biosciences), using lipofectamine LTX (Thermofisher). HEK-293T media was collected 72 h post transfection, passed through a 45 nm syringe filter, and applied to neuronal media. Drug treatments were performed after 15–18 days in vitro in fresh medium containing Neurobasal-A supplemented with 2% B27. Neuronal cell viability was determined following manufacturer's protocol using CCK8 kit (Dojindo) in cortical neurons grown in 96-well plates, seeded at a density of 30,000 cells/well and grown for 15 days in vitro.

**Animals.** Animals were used in full compliance with National Institute of Health/Columbia University Institutional Animal Care and Use Committee guidelines. The animal protocol was approved by the Committee on the Ethics of Animal Experiments of Columbia University. *Pik3c3*flox/flox mice were a kind gift of Dr. Fan Wang (Duke University School of Medicine) and have been previously described[12,13]. Mice were crossed with transgenic mice expressing *Cre* recombinase under the promoter of *CaMKII* to conditionally knockout *Pik3c3* in excitatory forebrain neurons. At 2 and 3 months of age, mice were killed by cervical dislocation, brains immediately macro-dissected, frozen in liquid nitrogen, and stored at −80 °C. In all experiments, littermate *Pik3c3*flox/flox; *CaMKII-Cre* (*Pik3c3* cKO) of both sexes were compared to *Pik3c3*flox/flox (CTRL).

**Immunofluorescence and confocal microscopy.** Cultured cortical neurons were fixed in 2% paraformaldehyde (PFA, Alfa Aesar), 2% sucrose (Sigma-Aldrich) in culture media for 15 min and permeabilized with 0.05% saponin in phosphate-buffered saline (PBS, Boston Bioproducts) supplemented with 1% bovine serum albumin (BSA, Sigma-Aldrich). Primary and Alexa Fluor conjugated secondary antibodies (Thermofisher, Jackson Immunoresearch) were sequentially incubated for 1 h in the same buffer. Coverslips were mounted in ProLong Gold antifade mountant (Thermofisher). For immunohistochemistry, mice were transcardially perfused with PBS. Brains were dissected and fixed overnight in 4% PFA in PBS at 4 °C, followed by incubation in 30% sucrose (Sigma-Aldrich) for 48 h in PBS. Brains were sectioned in coronal planes with a cryostat (Leica Biosystems). Immunostaining was performed by blocking free-floating brain sections in 5% Donkey Serum (Thermofisher), 1% BSA and 0.2% Triton X-100 (Fisher Scientific) in PBS for 1 h. Primary antibodies were incubated overnight at 4 °C and secondary antibodies for 2 h at room temperature, both in blocking solution. Slices were mounted in ProLong Gold antifade mountant (Thermofisher). Confocal stacks and super-resolution images were acquired using a Zeiss LSM 800 confocal microscope equipped with Airyscan module (Zeiss). Fluorescence was collected with ×40 and ×63 plan apochromat immersion oil objectives. Extraction of single z-frame and maximum intensity projections were performed with ImageJ software. Colocalization was calculated with JACoP plugin of ImageJ and is expressed as Manders' coefficient. Object number, size and intensity were detected and quantified with the ICY software. Each experiment was independently repeated at least three times, unless indicated otherwise.

**Conventional electron microscopy.** Cultured cortical neurons were fixed with 2.5% glutaraldehyde in 0.1 M cacodylate buffer (Electron Microscopy Sciences) for 24 h and processed as described previously[15]. Ultrathin sections were prepared with an EM UC6 ultracryomicrotome (Leica). Exosomes purified from mice brain (see below) were suspended in PBS and fixed with a mixture of 2% PFA/0.065%

glutaraldehyde in 0.2 M PBS at 4 °C. The suspension was loaded onto formvar–carbon-coated EM grids (Electron Microscopy Sciences) and fixed a second time. Samples were contrasted and embedded in a mixture of uranyl acetate and methylcellulose (Electron Microscopy Sciences). Images were acquired using a Philips CM-12 electron microscope (FEI) and digital acquisitions made with a Gatan (4 k × 2.7 k) digital camera (Gatan).

**Protein biochemistry and immunoblotting**. Cells were washed in PBS and scraped in Radioimmunoprecipitation assay buffer (RIPA) (Pierce) supplemented with protease and phosphatase inhibitor cocktail (Roche). Homogenates were centrifuged for 10 min at 16,000 × g at 4 °C. Subcellular fractionation of particulate and soluble fractions was performed as previously described[70]. Supernatants were processed for protein dosage (BCA, Pierce) and samples diluted to equal concentration. SDS–PAGE was carried out per manufacturer's protocol (Thermofisher). Samples (20–40 µg protein) were prepared with NuPage LDS sample buffer and NuPage reducing reagent and loaded in NuPAGE 4–12% Bis-Tris gels; separation was carried out using MES running buffer (Thermofisher). Wet transfer was performed on 0.22 µm nitrocellulose membranes (Amersham) at 80 V for 1h45 min at 4 °C using 0.05% SDS tris-glycine buffer (Boston BioProducts). Membranes were blocked with 5% BSA for 45 min at room temperature. Primary antibodies were diluted in blocking buffer and incubated overnight at 4 °C; HRP-conjugated secondary antibodies were incubated for 90 min at room temperature. Membrane development was performed with Immobilon Western Chemiluminescent HRP Substrate (EMD Millipore) and the chemiluminescent signal was imaged with ImageQuant LAS4000 mini (GE Healthcare). Quantification was performed with ImageJ. Analysis of APP-CTFs was done using NuPage 10–20% Tricine gels (Thermofisher). For analysis of mouse brain tissue, samples were homogenized in 10 volumes of RIPA supplemented with protease and phosphatase inhibitor cocktail (Roche). Homogenates were centrifuged at 16,000 × g for 15 min and supernatants processed as described above. Uncropped blots are shown in Supplementary Fig. 9.

**Aβ measurements**. Cortical neurons were grown 15 to 18 days in vitro. Neurons were placed in fresh culture medium containing Neurobasal-A and 2% B27 and treated for 24 h. Conditioned media was collected, treated with AEBSF protease inhibitor (1 mM, Thermofisher), centrifuged at 2000 × g for 5 min and stored at −80 °C. Levels of Aβ40 and Aβ42 were measured using V-PLEX Aβ Peptide Panel 1 (4G8) kit (MesoScaleDiscovery, MSD) following the manufacturer's protocol. Mouse hippocampi were homogenized in 10 volumes of tissue homogenization buffer (250 mM sucrose, 20 mM Tris base, 1 mM EDTA, 1 mM EGTA) and diluted 1:2 in 0.4% diethanolamine, 100 mM NaCl to extract soluble β-amyloid. Samples were ultracentrifuged for 60 min, at 4 °C and 100,000 × g using a TLA55 rotor (Beckman Coulter) and equilibrated with Tris-HCl (0.05 M). Murine Aβ40 and Aβ42 were measured using Aβ40 and Aβ42 Mouse ELISA Kits (Thermofisher) following the manufacturer's protocol. In both cases, samples were measured in duplicates and values were normalized to protein concentration of cell or brain lysates, respectively.

**Exosome isolation**. Primary cortical neurons were grown in 100 mm dishes or 6-well plates at a density of 100,000 cells/cm². Conditioned media was collected 24 to 36 h after treatment and exosomes isolated by differential centrifugation as previously described[34,39], with minor modifications. Briefly, samples were cleared of cellular debris at 2000 × g for 20 min, filtered through a 0.2 µm PES filter (Worldwide Life Sciences) and ultracentrifuged for 90 min at 4 °C and 100,000 × g using a Sw41 or TLA55 rotor (Beckman Coulter). Exosome pellets were washed in PBS and ultracentrifuged for 90 min at 4 °C and 100,000 × g. Pellets were resuspended in RIPA buffer for Western blotting or in PBS for lipid analysis (see below). Equal volumes of exosome resuspension were loaded on SDS–Page. For analysis of N2a cell-derived exosomes, FBS was exosome-depleted by ultracentrifugation for 18 h, at 4 °C and 100,000 × g. Cells were grown in 10% FBS and media was processed as described above. Purification of exosomes was confirmed by comparison of exosomal and non-exosomal markers levels in lysates and EVs. Brain exosomes were isolated as previously described[42,44] with slight modifications. Briefly, cortices were minced and digested in hibernate A and papain for 20 min at 37 °C (ThermoFisher). Digestion was halted with the addition of excess hibernate A containing protease and phosphatase inhibitors (Roche). The tissue was homogenized gently with a serological pipette and cells were removed by centrifugation for 10 min at 4 °C and 300 × g. The supernatant was passed through a 40 µm strainer (BD Bioscience) followed by a 0.2 µm PES filter and subjected to serial centrifugations for 10 min at 2000 × g and 30 min at 10,000 × g, at 4 °C, to remove cellular debris. The resulting supernatant was centrifuged for 70 min at 4 °C and 100,000 × g, washed with PBS and centrifuged again. Pellets were suspended in 3 mL of 40% Optiprep (Sigma-Aldrich), overlaid with 3 mL of 20, 10, and 5% Optiprep solutions and centrifuged for 18 h at 4 °C and 200,000 × g. One-ml fractions were collected, diluted in PBS, and centrifuged again for 70 min at 4 °C and 100,000 × g. Pellets were resuspended in RIPA for immunoblotting or PBS for EM and lipid analysis. Fraction density was calculated by diluting a sample in three volumes of 0.25 M sucrose and measuring optical density at 340 nm (Axis-Shield Density Gradient Media). Validation of exosome purification was performed in wild-type C57BL/6

mice by buoyancy, EM and enrichment of exosome markers ALIX and Flotillin-1. Fractions containing exosomes were pooled together for downstream analyses.

**Lipid analysis**. Lipid extracts were prepared using a modified Bligh/Dyer extraction as previously described[15]. Briefly, cells or mice hippocampi were resuspended and homogenized in a solution of methanol:chloroform (2:1) and lipids extracted using chloroform:KCl (3:2, 1 M). Extracted lipids were dried under vacuum and stored at −80 °C. Extracts were spiked with appropriate internal standards and analyzed by LC–MS. For analysis of anionic phospholipids and phosphoinositide, extracted lipids were deacylated, and analyzed by anion-exchange high-performance liquid chromatography with suppressed conductivity[15]. Lipid levels are expressed as average Mol% of the total sum of moles of lipids detected. The nomenclature abbreviations are: FC, free cholesterol; CE, cholesterol ester; MG, monoacylglycerol; DG, diacylglycerol; TG, triacylglycerol; Cer, ceramide; dhCer; dihydroceramide; SM, sphingomyelin; dhSM, dihydrosphingomyelin; MhCer, monohexosylceramide; Sulf, sulfatide; LacCer, lactosylceramide; GM3, monosialodihexosylganglioside; PA, phosphatidic acid; PC. Phosphatidylcholine; PCe, ether phosphatidylcholine, PE, phosphatidylethanolamine, PEp, plasmalogen phosphatidylethanolamine, PS, phosphatidylserine; PI, phosphatidylinositol; PG, phosphatidylglycerol; BMP, bis(monoacyl)glycerol; LPC, lysophosphatidylcholine, LPCe, ether lysophosphatidylcholine; LPE, lysophosphatidylethanolamine; LPEp, plasmalogen lysophosphatidylethanolamine; LPI, lysophosphatidylinositol.

**Statistics**. Statistical analysis was performed using Prism software (Graphpad). All the data are given as mean ± s.e.m. for a given N of biological replicates. Results were pooled from independent experiments as indicated. No statistical method was used to determine sample size. Mice were randomized per litter with age-matched controls and results pooled from at least two distinct litters. No blinding was done for biochemical analyses. For comparison of two experimental conditions, two-tailed Student's t test was performed. One-way ANOVA followed by Holm–Sidak's multiple comparisons test was performed for analysis of additional experimental groups. Analysis of time-course experiments was performed with two-way ANOVA repeated measures followed by Holm–Sidak's multiple comparisons test. Exact p values are reported in Supplementary Table 1.

**Data availability**. The authors declare that all data supporting the findings of this study are available within the paper and its supplementary information files. Data are available from the corresponding author on request.

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

## Acknowledgements

We thank Fan Wang for the kind gift of the *Pi3kc3*flox/flox mice. We thank Basant Abdulrahman and Hermann Schaetzl for providing the gene-edited *Atg5* KO N2a cells. We are also grateful to Zhenyu Yue, Ralph Nixon, and Jean Gruenberg for the kind gift of anti-Atg14L, Cathepsin D, and BMP antibodies, respectively. We thank Thomas Südhof for sharing *Cre* recombinase lentiviruses. We thank the OCS Microscopy Core of New York University Langone Medical Center for the support of the EM work and Rocio Perez-Gonzalez and Efrat Levy of New York University for their support during optimization of the brain exosome isolation technique. We thank Elizabeta Micevska for the maintenance and genotyping of the animal colony and Bowen Zhou for the preliminary

lipidomic analysis of conditional *Pi3kc3* cKO mice. We also thank Rebecca Williams and Catherine Marquer for critically reading the manuscript. This work was supported by grants from the Fundação para a Ciência e Tecnologia (PD/BD/105915/2014 to A.M.M.); the National Institute of Health (R01 NS056049 to G.D.P., transferred to Ron Liem, Columbia University; T32-MH015174 to Rene Hen (Z.M.L.)). Z.M.L. and R.B.C. received pilot grants from ADRC grant P50 AG008702 to S.A.S.

## Author contributions

A.M.M., T.G.O., S.A.S., and G.D.P. designed the research. A.M.M. coordinated and carried out the bulk of the experiments. Z.M.L. performed the characterization of cKO neurons and mice. A.M.M., Z.M.L., Y.X., S.S., and R.B.C. designed and carried out the lipid biochemistry. J.N. purified and characterized exosomes from mice brain. S.S. performed the EM analyses. A.M.M. and G.D.P. wrote the manuscript. All co-authors edited the manuscript. G.D.P. conceived the project and supervised the study.

## Additional information

**Competing interests:** G.D.P. is a full-time employee of Denali Therapeutics Inc. The remaining authors declare no competing financial interests.

