## [Peer Review File · Nature Communications]

Editorial Note: Parts of this peer review file have been redacted as indicated to maintain confidentiality.

Reviewers' comments:

Reviewer #1 (Remarks to the Author):

This is an interesting and well done study, investigating the effects of vps34 on endolysosomal function, autophagy and EV release of APP CTFs. The authors find that VPS34 deficiency results in endolysosomal membrane damage, defects in autophagy and increased release of APP CTFs via neutral SMnase dependet EV subtypes which are enriched in BMP.

Also this work is highly relevant and the experiments are technically sound, I have several concerns:

Suppl Fig 1b: The pharmacological inhibition of VPS34 wiht VPS34iN increases levels of Beclin. Could this explain some of the results ? Have the authors recapitulated key in vitro findings with VPS34 knockdown instead ? I ask this question also in light of the partially different results obtained in the in vivo situation with cre mediated vps34 knockdown.

Fig 1a: It is difficult to appreciate the enlarged EEA1 endosomes in the VPS34iN1 condition. Would it be possible to find a better image ?

Suppl 2b " The remaining degradation was blocked by Baf1" I assume that this conclusion should be based on the comparision of VPS34 inhibition versus VPS34 inhibition +Baf (or have I misunderstood remaining degradation). In that case, I would expect the statistics to compare these two conditions.

Suppl 2d "Gamma secr inhibitor extended APP CTF halflife relative to Cyclohexamide alone, confirming that g secr cleavage is a major pathway for clearance of APP CTFs": Fig 1d compared to 1c seems to show the opposite.

Authors claim that VPS34 inhibition cuases decreased lysosomal degradation of APP CTFs. Do APP CTFs accumulate in lysosomes or ILVs during combined g-secretase+Baf treatment ?

Fig 3c: could authors also quantify galectin 3/lamp/flot colocalisation or provide better images ? they claim that damaged endolysosomes are not efficiently targeted to Lamp1 pos structures upon VSP34. This can not really be judged by the image provided

Fig 4,5, Suppl Fig 6: I think that a quantification of protein release with EVs should be done in all cases by calculating the ratio of protein in EV/lysate and normalize this to control conditions. Otherwise, it is hard to tell, whether an effect is based on differences in protein concentrations in the lysate and a statistical significance cannot be concluded from the histograms if fold changes in EVs are significant and also in lysates but a direct comparison of EV/lysate between 2 conditions could still be not significant. E.g. , in Fig. 5 there is a 15 fold enrichment of CTFs in lysates and also in EVs upon Bafilomycin treatment, but there may be no net increase if you quantify EV/lysate.

Fig 6 Here, I am missing the GW4869 condition alone. GW4869 has many additional effects (other than just inhibition of SMnase dep. EV release). Thus, I am wondering, whether it would also inhibit the release of polyubiquitinated proteins if given alone (a finding, which would be contradictory to previous literature)

Fig 7d Purification of brain derived EVs is very controversially discussed. At least the purity of this preparation should be shown by WBs with different markers of potential microsomal contaminations. The same holds true for the cell culture experiments where I have been missing these negative controls (at least it should be shown in one experiment that the quality of the EV preparation is good enough). For the brain derived EVs it would be helpful to provide an EM picture.

Why is there no APP full length present in the brain derived EVs ? It should... also according to the authors own findings from in vitro derived EVs.

Line 438: an "l" is missing in chloroquine

Discussion: Authors should discuss their work in the context of a previous publication by Guix et al., Mol Neurodegen. 2017, which also addresses endolysosomal/autophagy pathways, APP processing, release of APP CTFs by EVs and Abeta generation

Reviewer #2 (Remarks to the Author):

In their paper, Di Paolo and collaborators have analyzed the effects of VPS34 inhibition on endolysosomal and autophagic functions in neuronal cell lines and primary neurons. The rationale evolved from the previous work of the group on the implication of a dysregulated PI3P pathway in AD as well as on the emerging evidence of endolysosomal dysfunctions in early stages of neurodegeneration diseases. VPS34 is a key kinase playing dual roles in endolysosomal trafficking and autophagy through its functional inclusion in distinct regulatory/scaffolding complexes. In the current work they mainly used pharmacological inhibition of VPS34 and report that inactivation of VPS34 strongly affect lipid metabolism resulting in the accumulation of in particular ceramide, sphingomyelin and BMP species accompanied with the promoted secretion of BMP- and ceramide-enriched extracellular vesicles (EVs). In addition, overall these EVs contain as well flotillins and APP (FL and in particular CTFs), while resulting in decreased levels of secreted Abeta. The authors demonstrate elegantly that the APP-CTF accumulation originates from decreased lysosomal degradation and these effects can be countered by inhibitors of sphingomyelinase unequivocally linking the observations to aberrant sphingomyelin metabolism. Finally, they recapitulate part of these findings in vivo using a cKO model for VPS34. Overall the data are of high quality and the strong integration of very good cell biology, with biochemistry and lipidomic profiling makes their story appealing and of potential strong importance. Throughout reading I encountered however some problems with the interpretation of the data which are outlined below is some more detail. Provided that the authors can address

these caveats, a revised manuscript might be considered for publication.

Major compulsory points that require more scrutiny:

There are two major messages in the story, VPS34 deficiency or inactivation triggers a ceramide-driven pathway that shunts APP-CTFs to a subpopulation of EVs and overall this reveals the existence of a homeostatic response that may alleviate the effects of lysosomal dysfunction by secretion of EVs.

My first concern is that while the endolysosomal defects are majorly addressed using a combination of high quality imaging and biochemistry, the story falls short on the fact that all data related to APP and APP-CTFs are only biochemical. Nevertheless, throughout the text strong correlates are made between APP-CTFs accumulation and the observed morphological aberrancies however without actually showing that APP-CTFs indeed accumulate in the same flotillin-/BMP-/ceramide- positive organelles. This part of the story should be significantly improved.

Secondly, I struggle with correlating the imaging data with the interpretations and proposed mechanisms. Upon VPS34 inhibition the authors demonstrate, using high resolution imaging, the appearance of LAMP1-negative, LC3-II negative, but p62-positive organelles that co-stain for ubiquitinated cargo, galectin-3, flotillin-2 (figs 2-3). From this, the authors conclude that VPS34 inhibition blocks initiation of autophagy. However, both p62 and LC3-II are early markers of autophagy initiation and progression: the fact that organelles are positive for p62 might indicate that autophagy is initiated (this would also go in line with the Beclin increase monitored in Suppl Fig1). Moreover, most of the interpretation also clearly points to a defect in lysosomal function (degradation) indicating that autophagosomes might form but cannot fuse with existing lamp1-positive lysosomes. This is in line with their overall conclusion that VPS34 inh and the observed effects on lipid accumulation and APP-CTF build up is strictly related to the role of VPS34 in endolysosomal sorting. In addition, because of the co-localization of Gal3 on the p62-positive organelles, the authors suggest that these are the 'dysfunctional' or damaged (gal3 positive) lysosomes, but strangely they do not contain bona fide lysosomal markers like LAMP proteins. In several images it is also clear that the p62-positive organelles are mostly close to existing LAMP-positive organelles indicating a defect in docking/fusion events. However in the few EM-pictures none such close encounter is seen: instead electron-dense organelles are found close to 'empty' MVBs. If the p62-positive organelles are not related to autophagy, is it possible that they represent the 'empty' MVBs observed by EM (and thus not the electron dense organelles)?

Surprisingly, the authors are not really evaluating the distribution of MVB markers, like some tetraspanins. In addition, they show co-localization of BMP with lysosomes in control cells but not in VPS34 inhibited cells. Maybe visualization of the accumulating lipids (ceramide and LBPA) in control vs VPS34 inh cells, in correlation with an extended set of MVB markers might better elucidate the identity of these organelles. An additional control might be to consider inhibition of the ESCRT pathway in conjunction with VPS34 inhibition as the authors mention that the ceramide-pathway to ILVs is ESCRT independent.

Finally, on page 9 (line 204) the authors hypothesize that the decrease in PI3P, as seen in LSD and following VPS34 inhibition might result in secondary lipid dysregulation. However this is not further discussed. Can the authors speculate how this mechanistically could work? Further, they focus on the most obvious lipids that are accumulating (ceramides, sphingomyelins, BMPs). However, in different cellular models, other lipid species are

significantly downregulated, notably storage lipids and its precursor PA. Others, like lyso-species are higher in EVs of N2A and in hippocampus but lower in EVs of primary neurons. Do the authors have an interpretation for these other changes?

More detailed remarks that should be addressed:

- On few occasions the authors make the strong statement that (early) endosomal enlargement is a 'key endosomal phenotype of AD'. I would be more cautious as these observations are limited to only a few studies. Nevertheless, in neurodegeneration one might indeed focus more on endo-lysosomal transport defects instead of autophagy: this is reviewed recently in Peric et al (*Acta Neuropathologica*, 2015) and might be considered to include as a reference. Related to this, it surprises me that the EE enlargement is so strong, given that the provided confocal images are less convincing. It would be stronger to support this with EM images of EE (given that EM is performed on these cells, this should be feasible to quantify).
- The authors measure in some cases APP-FL but mostly restrict the interpretation to APP-CTF. While for instance in fig 1E, FL is not altered, it increases as well on EVs of primary neurons (4b). Given the variability between observed in effects in different cell lines, it should be more logic to include systematically the CTF/FL ratio. Furthermore, to exclude defects in BACE1 processing, measurements of sAPPbeta should be included.
- Suppl Fig3: panel b and c are inverted in the legends.
- Fig. 4b: Ponceau staining shows equal amount of protein for EVs of control and VPS34 inh neurons. It might suggest that there is not an increase in the number of EVs released, but the released EVs contain more APP/CTF, flotillin etc. Please clarify.
- p12, line 271: the authors refer to fig 4e and suppl fig 5b to show that Cer levels were increased in EVs of N2a but slightly decreased in EVs of primary neurons. The correct figure panel is however fig 4c.
- p15. In their last part the authors recapitulate PI3P-dependent endolysosomal dysfunction in vivo through the analysis of VPS34 cKO mice. It might be informative for the reader to include in suppl data some evidence for a neurodegeneration phenotype in these mice.
- p15, line 362. In EVs isolated from brain not flotillins but Alix is significantly increased: this is a very surprising finding as in cell lines, Alix is mostly not affected but flotillins are increased in EVs following VPS34 inhibition. One could argue a contribution of non-neuronal cells, but this mean that in these cells flotillins are significantly less present to mask their upregulation in neuronal EVs. How to explain this discrepancy?
- p16, line 383: the authors generalize their conclusion ('EVs as part of a homeostatic response counteracting LDSs') too much. They demonstrate that VPS34 inh promotes secretion of a subtype of EVs and that this is related to lysosomal storage defects, not lysosomal storage diseases. The extrapolation to disease models need to be implemented to support this statement.
- p21, line 493-500: please include few references underscoring the link that is made to immune cells, DAMPs etc.

Reviewer #3 (Remarks to the Author):

This manuscript describes the biological effect of inhibition of VPS34, a class III PI 3-kinase regulating endosomal trafficking, on endosomal dysfunction and subsequent accumulation of APP CTF, other endosomal molecules, and their enhanced release in exosomes. The study incorporates comparison of murine neuronal cell line N2a and primary cultured mouse neurons, and detailed characterization of endosomal accumulation of several key markers by high-resolution microscopic imaging techniques. The in vitro findings are partly verified by in vivo study using conditional neuron-specific knockout of VPS34 in forebrain. The study is significant by finding a novel VPS34-dependent pathway of exosome secretion of APP-CTF after endocytic dysfunction, the identification of the new lipid BMP, and the identification of their dependence in sphingomyelin metabolism to ceramide synthesis as determined by GW4869 and myriocin. These findings would have biological implication in generalized exosome secretion mechanism after endocytic dysfunction in many disease conditions, and BMP has a potential for a specific marker for the extracellular vesicles reflecting the intracellular dysfunction of endosome machinery in the affected cells. The study, however, has many data, which are inconsistent among experimental settings and most of them are not well discussed. These have to be clearly addressed.

Major points:

1. There is no introductory description of what p62 expression means in this manuscript. P62 first appears on Fig. 2a (line 185), which is one of key molecules, without description. Please address this in the introduction.
2. Clearly describe and discuss why Beclin1 expression is specifically increased in VPS34IN1-treated group (Fig. S1b) instead of describing that this treatment "did not cause a general downregulation."
3. It is difficult to agree that the diameter of EEA1+ puncta is increased by VPS34IN1 treatment (Fig. 1a), although the reduction of its intensity is agreeable. Immuno-EM of EEA1+ cells will be more conclusive for the morphological analysis.
4. In Fig. 1b, the size of Rab5+ puncta appears to be increased by VPS34IN1 treatment, which is not mentioned in Fig. 1a. Please discuss.
5. The conclusion of the last sentence (line 132-133) is not well supported, since these data are not directly relevant to AD. This has to be rewritten.
6. In Fig. 1d, there is no obvious increase of APP-CTF or APP-CTFbeta fragment in the WB images. This has to be replaced with more representative images. Please also discuss why the effect of VPS34IN1 treatment on APP processing is more significant in N2a cells over primary cultured neurons.
7. The LC3 signal in Fig. 2a-b is almost invisible in all 6 panels. MAP2 signal is also very weak. Ub signal in Fig. 2c is also invisible. Figure 2 images should be replaced with more visually understandable images.
8. In Fig. 3, galectin-3 is a well-accepted marker of phagocytic myeloid cells, and it is rarely detected in neurons. The co-localization of galectin-3 and flotilin-2 in Fig. 3d is also invisible and their quantification is in question. Suggest to delete Fig. 3b-d.
9. In Fig. 4-7, there are WB images of proteins in the purified EV in primary neurons, N2a cells and VPS34 cKO mouse brains. Although the data are of high quality, there are several inconsistent findings, which are not well discussed to understand the differences. For example, Flotilin-1/2 expression is increased by VPS34IN1 treatment in the EV isolated from

primary neurons and N2a cells (Fig. 4b,d) but not in EV isolated from VPS34 cKO mouse brain (Fig. 7d). On the other hand, another EV marker ALIX expression increased by VPS34IN1 treatment in the EV isolated from primary neurons (Fig. 4b) and VPS34 cKO mouse brain (Fig. 7d) but not in EV isolated from N2a cells (Fig. 4d). This indicates that none of the tested EV markers show consistent increase by inhibition or deletion of VPS34, and suggest that the group of EV affected by VPS34 inhibition is highly dependent on the conditions although they are of neuronal origin. This should be clearly discussed.

10. In addition, the expression of full-length APP in the EV is different in these experiments. It is increased by VPS34IN1 treatment in the EV isolated from primary neurons (Fig. 4b) but not in EV isolated from N2a cells (Fig. 4d), and its expression in VPS34 cKO mouse brain is not shown. There is no clear explanation of these inconsistent findings or experimental approach.

11. As for p62 in EV, there are repetitive discussion of the existence of p62 in the EV after VPS34 inhibition in primary cultured neurons and N2a cells (Fig. 4b,d), which suggests the shuttling of dysfunctional endolysosomes to EV as p62-containing complex. However, p62 expression is not shown in the EV fraction isolated from VPS34 cKO mice (Fig. 7d). This is a key conclusive data and should be presented.

12. In Fig. 6b, treatment of cells with nSMase2 inhibitor GW4869 would suppress the total number of EV secreted to the media. Please clarify how much is the input of the EV for each sample.

13. This reviewer disagrees with the conclusion that Atg5 KO has no effect on the secretion of APP-CTF in the EV fraction. Supplementary Fig. 6d shows enhanced accumulation of APP-CTF in ATG5 KO N2a cells, which appears to be more than wildtype N2a cells (lane 1) and is further accumulated by VPS34IN1 treatment (lane 4). This clearly shows that ATG5 deletion inhibit autophagic function and enhances the secretion of APP-CTF in the EV fraction. The authors show the intensity as fold increase of untreated cells, but this is not the correct analysis of band intensity. The band intensity should be quantified on the same SDS-PAGE after loading the same amount of protein from different cell types. The data presentation and interpretation should be thoroughly revised.

14. The Supplementary Fig. 4c does not have the methods to understand how BMP is stained for imaging, and does not appear to be co-localized with LAMP-1. This description (line 213-215) should be revised or replace the data with more representative image.

15. In general, mono-ubiquitination is necessary for the sorting of proteins to ILV via ESCRT machinery, but the protein is deubiquitinated before the insertion into the ILVs. This study repeatedly shows poly-ubiquitinated molecules in the EV fraction (Fig. 4b, 6b, 7d), which is odd but may represent non-ESCRT machinery for the insertion of these molecules to ILVs. Please cite at least one reference reporting the existence of poly-ubiquitinated molecules in the ILVs or exosomes to substantiate the finding.

16. Line 277-278 says "EVs containing BMP can be unambiguously defined as bona fide exosomes". This should be toned down since later in the discussion BMP is described as a unique marker under specific conditions associated with endolysosomal dysfunction (line 471-473).

Minor comments:

1. In Fig. 1c image, please clarify if the CatD refers to only processed CatD or both CatD and proCatD.

Response to reviewers' comments:

Reviewer #1 (Remarks to the Author):

This is an interesting and well done study, investigating the effects of vps34 on endolysosomal function, autophagy and EV release of APP CTFs.

We thank this reviewer for his/her positive comments.

The authors find that VPS34 deficiency results in endolysosomal membrane damage, defects in autophagy and increased release of APP CTFs via neutral SMnase dependent EV subtypes which are enriched in BMP.

Also, this work is highly relevant and the experiments are technically sound, I have several concerns:

1. Suppl Fig 1b: The pharmacological inhibition of VPS34 with VPS34iN increases levels of Beclin. Could this explain some of the results? Have the authors recapitulated key in vitro findings with VPS34 knockdown instead? I ask this question also in light of the partially different results obtained in the in vivo situation with cre mediated vps34 knockdown.

We thank this reviewer for raising this important question. We believe that the 50% increase in Beclin 1 levels observed upon VPS34iN1 treatment *in vitro* is not responsible for the observed phenotypes based on two new lines of evidence: first, conditionally knocking out *Pik3c3/Vps34* in neurons causes a downregulation of Beclin 1 in mouse hippocampi (see revised Figure 7b, with new Beclin 1 immunoblot performed on the same hippocampal extracts/blots) and yet recapitulates the main findings observed *in vitro* using VPS34iN1, including aberrant sphingolipid metabolism and accumulation of p62, poly-ubiquitinated proteins and APP-CTFs. Additionally, brain exosomes from the KO mice also exhibit higher levels of APP-CTFs and the lipid BMP, based on our new lipidomics results (see revised Figure 7g), consistent with the increase we have observed in exosomes produced by VPS34iN1-treated neurons or N2a cells. Second, we have conducted new experiments showing that lentiviral expression of Cre recombinase in *Pik3c3*^{flox/flox} primary cortical neurons also recapitulates some of the key phenotypes observed in VPS34iN1-treated neurons, including accumulation of p62, as seen by Western blotting and immunostaining; co-localization of ubiquitin and p62 and, also importantly, recruitment of galectin-3 to p62-positive structures. We have now added these new data in revised Supplementary Figures 3 and 4 and discuss them on page 9 (lines 191-194) and page 10 (line 228-230). Importantly, we show below that Cre-mediated *Vps34* ablation in cultured neurons downregulates Beclin 1 levels (n=2) (Figure 1 for Referees) similarly to what we have previously reported in *Pik3c3* KO MEFs (Devereaux et al. 2013). We now comment on this important point in the discussion from the revised MS, on page 18 (lines 419-422).

Figure 1. Cre-mediated VPS34 knock-out causes downregulation of Beclin1. Primary neurons derived from $pik3c3^{flox/flox}$ mice were infected with ΔCre or Cre lentivirus at 7 days *in vitro* and processed for western blot analysis 8 days post-infection. Bar graph indicates average protein levels, normalized to ΔCre -infected neurons (mean \pm range, N=2 biological replicates). For quantification of Vps34 knock-out efficiency, please see Supplementary Figure 3d.

2. Fig 1a: It is difficult to appreciate the enlarged EEA1 endosomes in the VPS34IN1 condition. Would it be possible to find a better image?

We have replaced the original pictures with an alternate set of pictures that better show the enlarged EEA1-positive endosomes in revised Figure 1a.

3. Suppl 2b "The remaining degradation was blocked by Baf1" I assume that this conclusion should be based on the comparison of VPS34 inhibition versus VPS34 inhibition +Baf (or have I misunderstood remaining degradation). In that case, I would expect the statistics to compare these two conditions.

We apologize for this omission and thank the reviewer for pointing it out. We have now added the statistical analysis for the direct comparison between Vps34 inhibition and combined treatment with BafA1 at 4 hr post-treatment, with a $p < 0.01$ after a one-way ANOVA using Holm-Sidak's post-test for multiple comparisons. Supplementary Figure 2b has been revised accordingly.

4. Suppl 2d "Gamma secr inhibitor extended APP CTF halflife relative to Cyclohexamide alone, confirming that g secr cleavage is a major pathway for clearance of APP CTFs": Fig 1d compared to 1c seems to show the opposite.

We believe that this reviewer may have been misled by the fact that the treatment duration shown in the x axis for APP-CTF levels is not the same for Supplementary Figure 2c and Figure 2d. Additionally, the y axes are different. With this in mind, we believe that our original interpretation was correct, namely that the γ -inhibitor XXI prolongs the half-life of APP-CTFs in the presence or absence of VPS34IN1. We hope the reviewer now agrees with our interpretation.

5. Authors claim that VPS34 inhibition causes decreased lysosomal degradation of APP CTFs. Do APP CTFs accumulate in lysosomes or ILVs during combined γ -secretase+Baf treatment?

We have previously reported that γ -secretase inhibition causes an increase in APP staining (using anti-cytoplasmic antibodies) throughout the cell, particularly in the endolysosomal system. We invite this reviewer to examine Figure 3 from our previous manuscript (Morel et al. 2013). Given this excess of intracellular APP-CTF observed with γ -secretase inhibition, we do not expect a discernable increase in APP-CTFs with the combined treatments (γ -secretase inhibition + BafA1), in line with the results shown in Supplementary Figure 2 suggesting that γ -secretase inhibition dramatically impairs APP-CTF clearance. However, given our results indicating that BafA1 treatment alone causes efficient sorting of APP-CTFs to exosomes, we hypothesize that APP-CTF accumulation in lysosomes or ILVs might be very transient, prior to release via exocytosis of multivesicular endolysosomal compartments. While it is theoretically an excellent idea to test how much APP-CTFs can accumulate in cells upon combined BafA1 treatment/ γ -secretase inhibition, we feel that it may be a bit tangential to this manuscript, which focuses primarily on Vps34 function. We hope the reviewer agrees with this view.

6. *Fig 3c: could authors also quantify galectin 3/lamp/flot colocalisation or provide better images? They claim that damaged endolysosomes are not efficiently targeted to Lamp1 pos structures upon VSP34. This cannot really be judged by the image provided.*

Following the reviewer's recommendation, we have replaced the original p62/Galectin-3/LAMP-1 confocal images with better images with slightly enhanced contrast. We have also added a linescan intensity analysis of a p62/galectin-3 structure in the vicinity of LAMP-1-positive membrane where it is obvious that p62/galectin-3 intensity lines peak outside of LAMP-1 intensity profile (see revised Figure 3c). In addition, we have included for the benefit of this reviewer four additional examples from conventional confocal microscopy showing that p62/galectin-3 puncta are largely excluded from the lumen of LAMP-1-positive structures (see Figure 2a for Referees). To support these conclusions, we have also included a distribution/linescan intensity profile from the co-staining of p62/flotillin-2/LAMP-1 after acquiring images with the Airyscan confocal microscope (see Figure 2b for Referees). Please note we also show that p62-positive structures are excluded from the lumen of LAMP-1-positive compartments after VPS34IN1 treatment in revised Figure 2b, even in the presence of BafA1, while these p62-positive structures accumulate luminally in LAMP-1 positive structures after BafA1-alone treatment).

Figure 2. Damaged endolysosomes are not efficiently incorporated in the lumen of LAMP-1-positive compartments. **a)** Additional representative pictures of cortical neurons treated with VPS34IN1 at 3 μ M for 24 hr and immunostained for MAP2, p62, galectin-3 and LAMP-1. Regular confocal insets highlight p62/Galectin-3 positive structures repeatedly seen in the periphery, but not lumen, of LAMP-1-positive compartments. **b)** Representative pictures of cortical neurons treated with vehicle or VPS34IN1 at 3 μ M for 24 hr and immunostained for MAP2, p62, flotillin-2 and LAMP-1. Airyscan insets highlight exclusion of p62/flotillin-2 structures from LAMP-1 lumen. Right panel, line intensity scan of P62/flotillin-2 structure and adjacent LAMP-1 compartment. Scale bar 10 μ m.

7. Fig 4,5, Suppl Fig 6: I think that a quantification of protein release with EVs should be done in all cases by calculating the ratio of protein in EV/lysate and normalize this to control conditions. Otherwise, it is hard to tell, whether an effect is based on differences in protein concentrations in the lysate and a statistical significance cannot be concluded from the histograms if fold changes in EVs are significant and also in lysates but a direct comparison of EV/lysate between 2 conditions could still be not significant. E.g., in Fig. 5 there is a 15 fold enrichment of CTFs in lysates and also in EVs upon Bafilomycin treatment, but there may be no net increase if you quantify EV/lysate.

We have followed the reviewer's excellent suggestion. In addition to the previous quantification of protein levels in EVs, we have added bar graphs showing the EV/Lysate ratio for full length APP (FL-APP) and APP-CTFs. Please see updated versions of Figures 4, 5, and 6 as well as Supplementary Figure 7.

8. *Fig 6 Here, I am missing the GW4869 condition alone. GW4869 has many additional effects (other than just inhibition of SMnase dep. EV release). Thus, I am wondering, whether it would also inhibit the release of polyubiquitinated proteins if given alone (a finding, which would be contradictory to previous literature)*

This reviewer is raising a very good point. Given that in our cell types analyzed, GW4869 dramatically decreases exosome release based on the loss of Alix in the EV preparation, we can safely state that the drug causes a decrease in polyubiquitinated proteins associated with EVs. However, we cannot conclude that the ubiquitin signals (which stems from a range of mono, multi- and polyubiquitinated proteins with different types of K links) serve as sorting signals for delivery into ILVs/exosomes via the canonical ESCRT pathway initiated by ESCRT0/Hrs. In fact, VPS34 inhibition has been shown by our lab (Morel et al. 2013) and many others before us to block the ESCRT pathway, which relies on Vps34's product phosphatidylinositol-3-phosphate (Schink et al. 2013). Therefore, while we do not believe our results contradict the literature, they certainly suggest that many ubiquitinated proteins (some of which could be cytosolic proteins) are sorted into ILVs/exosomes in a PI3P- and ESCRT-independent fashion. This is also consistent with other reports in the literature showing that ubiquitinated proteins can be sorted into ILVs and exosomes (Pisitkun et al. 2004; Huebner et al. 2016).

9. *Fig 7d Purification of brain derived EVs is very controversially discussed. At least the purity of this preparation should be shown by WBs with different markers of potential microsomal contaminations. The same holds true for the cell culture experiments where I have been missing these negative controls (at least it should be shown in one experiment that the quality of the EV preparation is good enough). For the brain derived EVs it would be helpful to provide an EM picture.*

This is a valid concern and we thank this reviewer for raising it. The protocols used for EV purification both *in vivo* and *in vitro* have been extensively validated (Perez-Gonzalez et al. 2012; Kowal et al. 2016; Sharples et al. 2008; Théry et al. 2006), although the consensus is that exosomes obtained from cultured cells are typically purer than those obtained from tissue, for obvious reasons. To address this reviewer's concern, we confirmed the quality of the EV preparation by showing absence of late endosome/lysosomal marker LAMP1, early endosome marker APPL1 and cytosolic protein GAPDH in EVs relatively to cell lysates (Supplementary Figure 5a). In addition, the proteins quantified in EVs (ALIX, flotillin-2 and APP-CTF) were found to be only enriched on the 100,000g pellet (100K) pellet and not the 2K or 10K pellets, which is diagnostic of small EVs (Kowal et al. 2016). Of note, we detected residual signal for flotillin-2 in 10K pellet but the presence of such vesicles is eliminated through filtration using 0.22um syringe filters as performed for all experiments shown in the manuscript (see Methods). This finding further suggests that the size of the EVs collected is that of typical exosomes (<200 nm). We also note that BMP is enriched on exosomes obtained both *in vitro* and *in vivo* after Vps34 inhibition/ablation (see new data in revised Figure 7g and Supplementary Figure 8d). As explained in our manuscript, the enrichment of BMP, an ILV lipid, on EVs, lends support to the notion that the vesicles we have purified are largely exosomes.

Regarding EV purification from mice, we now include two EM pictures of EVs in revised Supplementary Figure 8 as well as the biochemical characterization of various protein markers from fractions collected after ultracentrifugation of EVs on an Optiprep (iodixanol) gradient. We found that ALIX and flotillin-1 were enriched in fractions 4-7, namely in the range of 1.074-

1.108 g/mL, as previously described in (Greening et al. 2015; Klingeborn et al. 2017). Moreover, Golgi and ER markers GM130 and Calnexin, resp., were enriched in fraction 8-10 and were thus not used for further applications (see revised Supplementary Figure 8b,c,d). We have now updated the Methods section on page 28 (lines 661-693) to summarize our additional efforts to vet the exosome purification protocol.

10. Why is there no APP full length present in the brain derived EVs ? It should... also according to the authors own findings from in vitro derived EVs.

We [redacted] have not been able to detect significant amounts of FL-APP in 15-20ug preparations of brain derived EVs, not only in the mice tested for the purpose of this manuscript but also other animal lines only expressing endogenous wild-type APP. Please note that the APP-CTF/FL-APP ratio is reportedly increased in EVs (Perez-Gonzalez et al. 2012). For the purpose of this revision, we show that EV-associated APP-CTF/FL-APP ratio is increased comparatively to cellular lysates using N2a cells transiently transfected with APP-GFP (see Figure 3a for Referees). Moreover, we show that EVs are enriched for mature, glycosylated FL-APP as noted in the band shift to heavier weight in EV preparation in comparison to lysates, both in primary neurons and in N2a cells (see Figure 3b for Referees). Not only this suggests a de-enrichment of FL-APP in EV preparations, but it also suggests that the APP species detected are of endosomal origin, requiring FL-APP maturation in the Golgi, transport to the plasma membrane via secretory pathway and internalization and sorting into ILVs before release on EVs (Haass et al. 2012).

Figure 3. Enrichment of APP-CTF/FL APP ratio in EVs. **a)** N2a cells were transiently transfected with APP-GFP for 24 hours and cell culture media was processed for purification of EVs. APP-GFP-AICD: APP-GFP APP Intracellular Domain (Morel et al. 2013). **b)** Comparison of FL APP migration pattern in SDS-Page between cell lysates and EVs purified from primary cortical neurons or N2a cells treated with VPS34IN1 at 3uM and 1uM for 24hr, respectively.

11. Line 438: an "l" is missing in chloroquine

We thank the reviewer for noting this typo, which has now been corrected.

12. Discussion: Authors should discuss their work in the context of a previous publication by Guix et al., *Mol Neurodegen.* 2017, which also addresses endolysosomal/autophagy pathways, APP processing, release of APP CTFs by EVs and Abeta generation.

We thank the reviewer for suggesting the inclusion of this paper, which is now cited and discussed on page 20 (lines 476-482).

Reviewer #2 (Remarks to the Author):

In their paper, Di Paolo and collaborators have analyzed the effects of VPS34 inhibition on endolysosomal and autophagic functions in neuronal cell lines and primary neurons. The rationale evolved from the previous work of the group on the implication of a dysregulated PI3P pathway in AD as well as on the emerging evidence of endolysosomal dysfunctions in early stages of neurodegeneration diseases. VPS34 is a key kinase playing dual roles in endolysosomal trafficking and autophagy through its functional inclusion in distinct regulatory/scaffolding complexes. In the current work they mainly used pharmacological inhibition of VPS34 and report that inactivation of VPS34 strongly affect lipid metabolism resulting in the accumulation of in particular ceramide, sphingomyelin and BMP species accompanied with the promoted secretion of BMP- and ceramide-enriched extracellular vesicles (EVs). In addition, overall these EVs contain as well flotillins and APP (FL and in particular CTFs), while resulting in decreased levels of secreted Aβeta. The authors demonstrate elegantly that the APP-CTF accumulation originates from decreased lysosomal degradation and these effects can be countered by inhibitors of sphingomyelinase unequivocally linking the observations to aberrant sphingomyelin metabolism. Finally, they recapitulate part of these findings in vivo using a cKO model for VPS34. Overall the data are of high quality and the strong integration of very good cell biology, with biochemistry and lipidomic profiling makes their story appealing and of potential strong importance. Throughout reading I encountered however some problems with the interpretation of the data which are outlined below in some more detail. Provided that the authors can address these caveats, a revised manuscript might be considered for publication.

Major compulsory points that require more scrutiny:

1. There are two major messages in the story, VPS34 deficiency or inactivation triggers a ceramide-driven pathway that shunts APP-CTFs to a subpopulation of EVs and overall this reveals the existence of a homeostatic response that may alleviate the effects of lysosomal dysfunction by secretion of EVs. My first concern is that while the endolysosomal defects are majorly addressed using a combination of high quality imaging and biochemistry, the story falls short on the fact that all data related to APP and APP-CTFs are only biochemical. Nevertheless, throughout the text strong correlates are made between APP-CTFs accumulation and the observed morphological aberrancies however without actually showing that APP-CTFs indeed accumulate in the same flotillin-/BMP-/ceramide- positive organelles. This part of the story should be significantly improved.

We are very thankful of the overall positive comments regarding our work. We believe in the correlation of the APP-CTF biochemical data and lipid analysis as samples for protein quantification and lipid quantification have been processed in parallel in order for us to be able to analyze the same exosomal vesicle population.

In an attempt to address this concern, we have immunostained cortical neurons treated with vehicle or VPS34IN1 with antibodies to APP C-terminus (C1/6.1) and p62 (see revised Supplementary Figure 4d). We have focused on p62 because it shows the most robust staining pattern upon Vps34 inhibition and the antibody is of distinct origin (Guinea pig vs mouse, while anti-BMP and anti-flotillin-2 antibodies used in this manuscript are both of mouse origin). In line with our previous manuscript (Morel et al. 2013), few APP/APP-CTF are seen as distinct puncta in the somatodendritic compartment of cortical neurons. Importantly, we found little to no co-localization between APP/APP-CTFs and p62, although some proximity was seen between these two proteins after Vps34 inhibition. We speculate that the increased secretion of APP-CTFs in the form of exosomes is precisely what prevents their dramatic accumulation intracellularly.

2. Secondly, I struggle with correlating the imaging data with the interpretations and proposed mechanisms. Upon VPS34 inhibition the authors demonstrate, using high resolution imaging, the appearance of LAMP1-negative, LC3-II negative, but p62-positive organelles that co-stain for

ubiquitinated cargo, galectin-3, flotillin-2 (figs 2-3). From this, the authors conclude that VPS34 inhibition blocks initiation of autophagy. However, both p62 and LC3-II are early markers of autophagy initiation and progression: the fact that organelles are positive for p62 might indicate that autophagy is initiated (this would also go in line with the Beclin increase monitored in Suppl Fig1). Moreover, most of the interpretation also clearly points to a defect in lysosomal function (degradation) indicating that autophagosomes might form but cannot fuse with existing lamp1-positive lysosomes. This is in line with their overall conclusion that VPS34 inh and the observed effects on lipid accumulation and APP-CTF build up is strictly related to the role of VPS34 in endolysosomal sorting.

There is a large body of evidence indicating that phosphatidylinositol-3-phosphate (PI3P), the product of Vps34, is critical for autophagy initiation (Dall'Armi et al. 2013). p62 serves as an autophagy adaptor binding ubiquitinated cargoes during functional autophagy and it is known to both accumulate and aggregate when autophagy initiation (*i.e.*, autophagosome formation) is blocked (e.g., after silencing ATG9 and FIP200 (Kishi-Itakura et al. 2014)). Given that our results from Figure 2 and Supplementary Figure 3b suggest that Vps34 inhibition prevents formation of LC3-positive puncta (by immunofluorescence) and LC3 lipidation (by Western blot analysis), we have updated the manuscript to clarify that Vps34 impairment specifically blocks autophagy through inhibition of autophagosome formation (page 8, line 172-173). Moreover, we believe this phenotype is independent of Beclin 1 increase because our new data show that p62 accumulation and co-localization with ubiquitin or galectin-3 is recapitulated in Vps34 KO primary cortical cultures, where Beclin 1 protein levels are decreased (please see updated Supplementary Figure 3d,e, Supplementary Figure 5 and our response to reviewer 1's point #1).

3. In addition, because of the co-localization of Gal3 on the p62-positive organelles, the authors suggest that these are the 'dysfunctional' or damaged (gal3 positive) lysosomes, but strangely they do not contain bona fide lysosomal markers like LAMP proteins. In several images, it is also clear that the p62-positive organelles are mostly close to existing LAMP-positive organelles indicating a defect in docking/fusion events. However, in the few EM-pictures none such close encounter is seen: instead electron-dense organelles are found close to 'empty' MVBs. If the p62-positive organelles are not related to autophagy, is it possible that they represent the 'empty' MVBs observed by EM (and thus not the electron dense organelles)?

We thank the reviewer for highlighting this critical point and suggesting an alternate interpretation for our data. In general, we fully agree that the low colocalization between galectin-3 and LAMPs in VPS34IN1-treated neurons is inconsistent with lysosomal damage *per se*. In fact, proximity of galectin-3 to LAMPs rather than luminal staining of galectin-3 suggests that other compartments, likely early-to-late endosomes, may be the damaged entities. For that reason, the term lysophagy (*i.e.*, the autophagy of lysosomes) we have used in the text may therefore be inappropriate, although we note that Maejima *et al.* also show proximity to but not luminal sorting of galectin-3 into LAMP-1 compartments upon lysosome rupture in their seminal lysophagy paper (Maejima et al. 2013). It is also possible that upon rupture, these lysosomal structures may lose common membrane organelle markers, like LAMPs, perhaps via degradation by cytosolic proteases. In our experimental setting, however, we believe that the damaged organelles are of endosomal origin, as these are also flotillin-2 positive. As a result of this helpful and fair criticism, we have now rephrased the text and specifically refer to endolysosomal membrane damage, which is more accurate than "lysosomal damage".

Regarding the second part of the comment, we speculate that the p62-positive structures are the electron-dense membrane enclosed vesicles because of their appearance only after VPS34IN1 treatment and much smaller size than the 'empty' MVBs. We also suggest the empty MVBs to be LAMP1-positive given these are also enlarged upon VPS34IN1 treatment in confocal images (Figure 3c).

4. Surprisingly, the authors are not really evaluating the distribution of MVB markers, like some tetraspanins.

To address this concern, we have transiently transfected N2a cells with CD63-GFP and collected EVs after treatment with VPS34IN1 (see Figure 4 for Referees). We also took this opportunity to analyze the abundance of ESCRT0 and ESCRT1 proteins Hrs and Tsg101, resp. We did not observe any significant increase in any of the markers tested, while the experiment was internally controlled with the enrichment of APP-CTFs after Vps34 inhibition. We believe that secreted tetraspanin levels may reflect the total number of EVs and therefore does not exclude the enrichment of other EV-associated proteins per vesicle, such as flotillins and APP-CTFs. We now briefly mention these results on page 12 (Line 265-268).

Figure 4. EV secretion of multivesicular body (MVB)-associated markers. a) Western blot analysis of EVs collected from N2a cells transiently transfected with CD63-GFP, treated with vehicle or VPS34IN1 at 1uM for 24hr. EV protein levels were normalized to cell lysate protein concentration and cellular levels of each protein. Bar graph indicates relative protein levels normalized to vehicle (mean \pm SEM, N=3 biological replicates).

5. In addition, they show co-localization of BMP with lysosomes in control cells but not in VPS34 inhibited cells. Maybe visualization of the accumulating lipids (ceramide and LBPA) in control vs VPS34 inh cells, in correlation with an extended set of MVB markers might better elucidate the identity of these organelles.

This is a great point and we have now addressed this by conducting immunofluorescence experiments in N2a cells treated with VPS34IN1. We did not observe major differences in anti-BMP stainings between VPS34IN1- and vehicle-treated cells, consistent with the lack of major alterations of BMP levels in cells (as opposed to the dramatic changes observed in exosomes). As a positive control, we found that p62 puncta accumulate in the proximity of LAMP-1 positive compartments in VPS34IN1-treated N2a cells, similar to what we have observed in primary cortical neurons. We show the data in revised Supplementary Figure 4c and modified the text accordingly on page 10 (lines 208-210). Of note, we have not been able to successfully stain ceramide with an antibody, despite multiple attempts in the laboratory.

6. *An additional control might be to consider inhibition of the ESCRT pathway in conjunction with VPS34 inhibition as the authors mention that the ceramide-pathway to ILVs is ESCRT independent.*

This is an interesting suggestion from the reviewer. However, we and others have shown conclusively that Vps34-derived PI3P is critical to initiate the ESCRT pathway, simply because the early component of the ESCRT pathway, ESCRT0/Hrs, is recruited to endosomes precisely via interaction of its FYVE domain with PI3P. While we invite this reviewer to see the supporting evidence in our previous manuscript Morel *et al.* (e.g., Suppl. Figures S5, S6 and S7), there is also seminal work from Stenmark, Emr *et al.* that have shown this in multiple studies (Raiborg *et al.* 2013; Schink *et al.* 2013). Additionally, we have included in the text (page 20, lines 472-475) new data showing that a mutant of APP-GFP lacking all 5 ubiquitination sites in its C-terminus (via lysines-to-arginine mutations) (see (Williamson *et al.* 2017) for characterization of this mutant) is sorted into exosomes as efficiently as APP WT when N2a cells are treated with VPS34IN1, further indicating that Vps34 inhibition triggers an exosome release pathway that is independent of ESCRT.

7. *Finally, on page 9 (line 204) the authors hypothesize that the decrease in PI3P, as seen in LSD and following VPS34 inhibition might result in secondary lipid dysregulation. However, this is not further discussed. Can the authors speculate how this mechanistically could work?*

This is a fantastic question and we could have done a better job explaining our thoughts in the manuscript. To answer it, we invite this reviewer to check Box 3 from a beautifully-written review by Settembre, Ballabio and colleagues (Settembre *et al.* 2013). In essence, it is well established that mutations in lysosomal enzymes, including lipases like glucocerebrosidase, cause enzyme substrate accumulation, which, in turn, perturb the homeostasis of lysosomes through various mechanisms (ion dyshomeostasis, changes in lysosomal membrane composition, alteration of trafficking pathways and delivery of hydrolases to lysosomes, etc...). In the case of Vps34 inhibition, we are probably dealing with pleiotropic effects, ranging from retromer mistrafficking and alteration of mannose-6-phosphate receptor transport to aberrant hydrolase maturation (see for instance Figure 1c for Cathepsin D). In light of this comment, we elaborate on this point on page 18 (line 416-431) of the revised manuscript.

Box 3 | Mechanisms of lysosomal storage diseases (LSDs)

LSDs are a group of rare and recessively inherited metabolic dysfunctions with an overall incidence of 1 in 5000. LSDs are caused by mutations of genes encoding proteins that localize to the lysosomal lumen, lysosomal membrane or other cellular compartments that contribute to lysosomal function. These disorders are characterized by the progressive accumulation of material that has not been degraded in the lysosomes of most cells and tissues. Approximately 60 different types of LSDs have been recognized. Historically, LSDs have been classified on the basis of the type of material that accumulates in the lysosomes, such as mucopolysaccharides, sphingolipids, glycoproteins, glycogen and lipofuscins. LSDs often show a multisystemic phenotype that is associated with severe neurodegeneration, mental decline, cognitive problems and behavioural abnormalities. Other tissues that are commonly affected are bone and muscle. Cell and tissue pathology are the result of a complex series of pathogenic cascades that occur downstream of lysosomal dysfunction. The figure illustrates the main steps underlying LSD pathogenesis. Mutations in genes that are important for lysosomal function result in the accumulation of specific substrates that have not been degraded in the lysosome (primary storage). This leads to the accumulation of additional lysosomal substrates (secondary storage) due to a blockage in lysosomal trafficking. Excessive lysosomal storage has a broad impact on lysosomal function by causing defects in Ca^{2+} homeostasis, signalling abnormalities and lysosomal membrane permeabilization. In addition, lysosomal dysfunction is associated with autophagy impairment, due to defective fusion between lysosomes and autophagosomes. This causes the accumulation of autophagic substrates such as aggregate-prone proteins and dysfunctional mitochondria (tertiary storage), which contributes to neurodegeneration. GAGs, glycosaminoglycans.

8. Further, they focus on the most obvious lipids that are accumulating (ceramides, sphingomyelins, BMPs). However, in different cellular models, other lipid species are significantly downregulated, notably storage lipids and its precursor PA. Others, like lyso-species are higher in EVs of N2A and in hippocampus but lower in EVs of primary neurons. Do the authors have an interpretation for these other changes?

This is a great question from the reviewer and the short answer is "no", unfortunately. We have set up our lipidomics core several years ago and acquired a wealth of data in various settings, publishing many research papers. When analyzing about 30 lipid subclasses and hundreds of lipid species by LCMS, we typically find many that change. Complicating the interpretation of the data, each lipid is often regulated by multiple enzymes. Our strategy to gain more insights into the physiological and pathophysiological significance of lipid changes is to focus on pathways that can be easily manipulated by pharmacological or molecular genetic tools (although this step requires prioritization of lipid hits to pursue) and determine whether those manipulations alter functional outcomes. We also increasingly combine lipidomics with transcriptomic analyses, in the hope that mRNA changes may inform us on specific enzymes or pathways to further test or characterize.

9. More detailed remarks that should be addressed:

On few occasions the authors make the strong statement that (early) endosomal enlargement is a 'key endosomal phenotype of AD'. I would be more cautious as these observations are limited to only a few studies. Nevertheless, in neurodegeneration one might indeed focus more on endo-lysosomal

transport defects instead of autophagy: this is reviewed recently in Peric et al (Acta Neuropathologica, 2015) and might be considered to include as a reference.

We agree with the reviewer and thank him/her for this suggestion. We have revised our conclusion from Result section number 1 on page 6 (line 125-126). We have also added and discussed the Peric reference in the text on page 17 (line 400).

10. Related to this, it surprises me that the EE enlargement is so strong, given that the provided confocal images are less convincing. It would be stronger to support this with EM images of EE (given that EM is performed on these cells, this should be feasible to quantify).

In response to this comment and reviewer 1's point #2, we have included better pictures of the EEA1 staining in revised Fig. 1a. We hope that the reviewer can better appreciate the enlargement of the endosomal structures. Given that the EEA1-positive endosomal enlargement has been reported in several cell types both at the light and electron microscopic level (Morel et al. 2013; Devereaux et al. 2013; Futter et al. 2001; Jiang et al. 2010; Cossec et al. 2012), we felt that it was unnecessary to perform a time consuming morphometric analysis of ultrastructural analysis, and rather focus on strengthening the most novel aspects of our paper. We hope this reviewer will be clement enough to accept our explanation.

11. The authors measure in some cases APP-FL but mostly restrict the interpretation to APP-CTF. While for instance in fig 1E, FL is not altered, it increases as well on EVs of primary neurons (4b). Given the variability between observed in effects in different cell lines, it should be more logic to include systematically the CTF/FL ratio. Furthermore, to exclude defects in BACE1 processing, measurements of sAPPbeta should be included.

This is an excellent suggestion, partially brought up by reviewer 1 in point #7. We now provide the EV/Lysate ratios for FL-APP and APP-CTFs in Figures 4, 5, 6 and Supplementary Figure 6. Regarding the APP/APP-CTF ratio, we have shown in response to reviewer 1's point #10 that APP-CTFs/FL-APP is increased in EVs.

Regarding the contribution of BACE1 processing to altered APP metabolism caused by Vps34 inhibition, we were not able to detect sAPP β levels in the media of primary neurons. As a reminder, we performed all of our experiments studying endogenous, wild-type murine APP for which sAPP β levels are particularly challenging to detect, comparatively to studies overexpressing human WT or mutated APP. Nevertheless, we believe that an increase in BACE1-mediated processing of APP is unlikely to account for the APP-CTF increase observed upon Vps34 inhibition, given that levels of A β 40 and A β 42 are both decreased. Additionally, both APP-CTF α and APP-CTF β are similarly increased upon Vps34 inhibition, suggesting impaired clearance/processing downstream of α/β -secretase cleavage instead.

12. Suppl Fig3: panel b and c are inverted in the legends.

We thank the reviewer for noting this mistake, which has now been corrected.

13. Fig. 4b: Ponceau staining shows equal amount of protein for EVs of control and VPS34 inh neurons. It might suggest that there is not an increase in the number of EVs released, but the released EVs contain more APP/CTF, flotillin etc. Please clarify.

This is an excellent point raised by the reviewer. To precisely determine the number and amount of exosomes released, we would have to resort to the use of a nanotracker instrument as in (Guix et al. 2017) which unfortunately is not available to us at Columbia University. To the best of our knowledge, we believe that the EV number is not significantly affected after Vps34 inhibition, particularly in N2a cells given the similar levels of EV-associated ALIX, Tsg101, Hrs and CD63-GFP. As mentioned in response to reviewer 2's point #4, we still believe this does not exclude the enrichment of proteins such as flotillins and APP-CTFs in EVs, likely as a subpopulation of these secreted vesicles. We have clarified this ambiguity in the revised manuscript on page 12 (Line 265-268).

14. - p12, line 271: the authors refer to fig 4e and suppl fig 5b to show that Cer levels were increased in EVs of N2a but slightly decreased in EVs of primary neurons. The correct figure panel is however fig 4c.

We thank again the reviewer for noting this mistake, which has now been corrected.

15. - p15. In their last part the authors recapitulate PI3P-dependent endolysosomal dysfunction *in vivo* through the analysis of VPS34 cKO mice. It might be informative for the reader to include in suppl data some evidence for a neurodegeneration phenotype in these mice.

This is a great suggestion. We have collected data on this topic, which we had originally intended for another manuscript. However, we agree it is important to document the extent of neurodegeneration in the mouse model we have used in this manuscript. We have now added new data in revised Figure 7a and Supplementary Figure 8a showing that there is no significant neuronal death at 2 months of age based on MAP2 staining of areas where *CAMKIIa-Cre* expression occurs (e.g. CA1 in the hippocampus) (Wang et al. 2011). Instead, massive loss of neurons is obvious at 3 months of age in the *Vps34* cKO mouse where a dramatic loss of MAP2 immunoreactivity is detected in both hippocampus and cortex, as well as thinning of cortical layers (Supplementary Figure 8a), as also reported by others characterizing the same mice (Wang et al. 2011). We discuss the data on page 15 (line 350-354) in the revised manuscript.

16. - p15, line 362. In EVs isolated from brain not flotillins but Alix is significantly increased: this is a very surprising finding as in cell lines, Alix is mostly not affected but flotillins are increased in EVs following VPS34 inhibition. One could argue a contribution of non-neuronal cells, but this mean that in these cells flotillins are significantly less present to mask their upregulation in neuronal EVs. How to explain this discrepancy?

This is a sharp observation from the reviewer and we have no clear explanation for it, although we can speculate. As pointed out by the reviewer, the EVs purified from the brain reflect the contribution of many cell types, including neurons, astrocytes, microglia and potentially also oligodendrocytes. While we selectively ablate *Vps34* from pyramidal neurons with *CamkIIa-Cre* and can safely conclude that changes in brain EVs by definition originate from those neurons, EV composition changes can also reflect contributions from these other cell types. Additionally, while EVs produced *in vitro* are typically accumulating in the media, there is a lot of evidence that EVs produced *in vivo* are also taken up by cells as part of cell-cell communication. This significantly complicates the interpretation of the results. However, the finding that are most relevant to this paper is that APP-CTFs and BMP levels (as shown in our new results in revised Figure 7g) are increased on purified EVs upon *Vps34* inhibition, both *in vitro* and *in vivo*. We have updated the text in page 21 (line 503-508) as to clarify this discrepancy.

17.- p16, line 383: the authors generalize their conclusion ('EVs as part of a homeostatic response counteracting LDSs') too much. They demonstrate that VPS34 inh promotes secretion of a subtype of EVs and that this is related to lysosomal storage defects, not lysosomal storage diseases. The extrapolation to disease models need to be implemented to support this statement.

We agree with this reviewer and have now toned down this statement, having no evidence that our findings are directly related to a *bona fide* LSD. However, we note that *Vps34* overexpression in myoblasts from patients with Danon disease was shown to alleviate the lysosomal/autophagy defects in this LSD primarily affecting the skeletal muscle (Nemazany et al. 2013), which certainly goes in this direction. Please find the revised text on page 17 (line 395-396).

18,- p21, line 493-500: please include few references underscoring the link that is made to immune cells, DAMPs etc.

We have now added the excellent review from Heneka et al. to support these potential implications (Heneka et al. 2014).

Reviewer #3 (Remarks to the Author):

This manuscript describes the biological effect of inhibition of VPS34, a class III PI 3-kinase regulating endosomal trafficking, on endosomal dysfunction and subsequent accumulation of APP CTF, other endosomal molecules, and their enhanced release in exosomes. The study incorporates comparison of murine neuronal cell line N2a and primary cultured mouse neurons, and detailed characterization of endosomal accumulation of several key markers by high-resolution microscopic imaging techniques. The in vitro findings are partly verified by in vivo study using conditional neuron-specific knockout of VPS34 in forebrain. The study is significant by finding a novel VPS34-dependent pathway of exosome secretion of APP-CTF after endocytic dysfunction, the identification of the new lipid BMP, and the identification of their dependence in sphingomyelin metabolism to ceramide synthesis as determined by GW4869 and myriocin. These findings would have biological implication in generalized exosome secretion mechanism after endocytic dysfunction in many disease conditions, and BMP has a potential for a specific marker for the extracellular vesicles reflecting the intracellular dysfunction of endosome machinery in the affected cells. The study, however, has many data, which are inconsistent among experimental settings and most of them are not well discussed. These have to be clearly addressed.

Major points:

1. *There is no introductory description of what p62 expression means in this manuscript. P62 first appears on Fig. 2a (line 185), which is one of key molecules, without description. Please address this in the introduction.*

We apologize for not making this clearer in the original manuscript. We have now provided some background to allow the reader to interpret better our experiments on page 8 (line 173-176) of the revised manuscript.

2. *Clearly describe and discuss why Beclin1 expression is specifically increased in VPS34IN1-treated group (Fig. S1b) instead of describing that this treatment “did not cause a general downregulation.”*

We thank the reviewer for pointing out that we have not described these data in the most accurate fashion. We have now corrected the statement by mentioning this increase and its unlikely role in the phenotypes we describe (see also our response to reviewer 1's point #1). Unfortunately, we have no clear explanation for why reducing the kinase activity of Vps34 may increase Beclin 1 levels. Because we do not believe it contributes to the phenotypes we describe, we have decided not to characterize it further.

3. *It is difficult to agree that the diameter of EEA1+ puncta is increased by VPS34IN1 treatment (Fig. 1a), although the reduction of its intensity is agreeable. Immuno-EM of EEA1+ cells will be more conclusive for the morphological analysis.*

Since this point as also raised by reviewers 1 and 2 (points #2 and #10, resp.), we invite this reviewer to read our responses. In essence, we have found better images that reflect our quantifications. Since we have already reported EEA1-positive endosomal enlargement upon Vps34 silencing in neurons and genetic ablation in MEFs (Morel et al. 2013; Devereaux et al. 2013), we have opted to focus on strengthening more novel aspects of our manuscript, rather than investing in a time-consuming immuno-EM analysis of the EEA1 compartment. We hope that this reviewer agrees with this justification.

4. In Fig. 1b, the size of Rab5+ puncta appears to be increased by VPS34IN1 treatment, which is not mentioned in Fig. 1a. Please discuss.

We apologize for omitting this information. We invite the reviewer to examine revised Supplementary Figure 1e which shows the average size of Rab5-positive puncta per cell. Given the marginal increase (4%) in puncta size after Vps34 inhibition, we decided to further characterize Rab5-puncta size distribution. Indeed, Vps34 inhibition caused an increase in the relative frequency of a larger subset of Rab5-positive endosomes (>0.23 μ m, 0.181 \pm 0.010 relative freq. for vehicle and 0.243 \pm 0.014 for VPS34IN1) at the cost of a reduction in smaller endosomes (<0.19 μ m, 0.524 \pm 0.010 relative freq. for vehicle and 0.462 \pm 0.012 for VPS34IN1). However, given the lower frequency of larger endosomes (~20-25%) comparatively to the other two groups of endosomes, these have a minor impact in the total sum of Rab5 size distribution. We have updated the manuscript accordingly on page 6 (lines 122-123).

5. The conclusion of the last sentence (line 132-133) is not well supported, since these data are not directly relevant to AD. This has to be rewritten.

We have rephrased the conclusion as suggested. Please find the revised version on page 6 (lines 125-126).

6. In Fig. 1d, there is no obvious increase of APP-CTF or APP-CTFbeta fragment in the WB images. This has to be replaced with more representative images. Please also discuss why the effect of VPS34IN1 treatment on APP processing is more significant in N2a cells over primary cultured neurons.

We thank this reviewer for pointing this out. We have addressed this point by replacing the original Western blot images with an alternate set in revised Figure 1d. We have also noticed and reported that the phenotype is more dramatic in N2a cells (page 7, line 151-153, and have no other explanation than the fact N2a cells are a cell line with fibroblast features that proliferate, unlike primary neurons. Overall, we feel confident that this finding occurs physiologically because it is observed in primary neurons as well as in mouse brain.

7. The LC3 signal in Fig. 2a-b is almost invisible in all 6 panels. MAP2 signal is also very weak. Ub signal in Fig. 2c is also invisible. Figure 2 images should be replaced with more visually understandable images.

We have replaced the original pictures with an alternate version with higher contrast evidencing all markers.

8. In Fig. 3, galectin-3 is a well-accepted marker of phagocytic myeloid cells, and it is rarely detected in neurons. The co-localization of galectin-3 and flotillin-2 in Fig. 3d is also invisible and their quantification is in question. Suggest to delete Fig. 3b-d.

We thank the reviewer for this fair criticism. We agree that galectin-3 is enriched in phagocytic myeloid cells comparatively with neurons (in fact, the gene encoding galectin-3, *Igals3*, has an 8-fold increase in transcript levels in microglia vs. neurons according to Brain RNA-Seq database developed by Ben Barres *et al.* (Zhang *et al.* 2014)). Despite the low expression levels, we were able to detect galectin-3 in primary neurons using a commercially available antibody for immunocytochemistry. While we use galectin-3 merely as a tool for the detection of endolysosomal membrane damage, which has been previously validated (Maejima *et al.* 2013; Aits *et al.* 2015; Paz *et al.* 2010; Bischoff *et al.* 2012; Papadopoulos *et al.* 2017), we do not exclude expression of other galectin isoforms or their accumulation in p62-positive structures in neuronal cells. As mentioned in response to reviewer 1's point #6, we have updated Figure 3b-d with optimized contrast and added linescan intensity plots (i) confirming luminal exclusion of p62/galectin-3 positive structures from LAMP-1 compartments and (ii) highlighting co-localization of p62/galectin-3/flotillin-2. Accordingly, we have updated the text in page 10 (line

219-222). We would like to keep these figures as we believe that endomembrane damage is a key phenotype of endolysosomal dysfunction induced by Vps34 inhibition.

9. In Fig. 4-7, there are WB images of proteins in the purified EV in primary neurons, N2a cells and VPS34 cKO mouse brains. Although the data are of high quality, there are several inconsistent findings, which are not well discussed to understand the differences. For example, Flotilin-1/2 expression is increased by VPS34IN1 treatment in the EV isolated from primary neurons and N2a cells (Fig. 4b,d) but not in EV isolated from VPS34 cKO mouse brain (Fig. 7d). On the other hand, another EV marker ALIX expression increased by VPS34IN1 treatment in the EV isolated from primary neurons (Fig. 4b) and VPS34 cKO mouse brain (Fig. 7d) but not in EV isolated from N2a cells (Fig. 4d). This indicates that none of the tested EV markers show consistent increase by inhibition or deletion of VPS34, and suggest that the group of EV affected by VPS34 inhibition is highly dependent on the conditions although they are of neuronal origin. This should be clearly discussed.

We hope we have addressed this concern in response to reviewer 2's point #16. We believe that the most relevant finding to this paper is that APP-CTFs and BMP are increasingly sorted to EVs as a result of endolysosomal dysfunction, particularly after Vps34 inhibition. We note, however, that the composition of EVs vary significantly between cell types (Kowal et al. 2016) and may account for the disparities in markers such as ALIX and flotillins across our study models. Accordingly, we have updated the text in page 21 (line 503-508) as to clarify this point.

10. In addition, the expression of full-length APP in the EV is different in these experiments. It is increased by VPS34IN1 treatment in the EV isolated from primary neurons (Fig. 4b) but not in EV isolated from N2a cells (Fig. 4d) and its expression in VPS34 cKO mouse brain is not shown. There is no clear explanation of these inconsistent findings or experimental approach.

We invite the reviewer to examine revised Figure 4d, which confirms that FL-APP is also significantly increased in EVs derived from VPS34IN1-treated N2a cells in comparison to controls. Concerning FL-APP levels in mouse brain EVs, we refer the reviewer to our response to reviewer 1's point #10. Briefly, we and our collaborators have not been able to detect significant amounts of endogenous FL-APP in exosomes derived from the brain of wild-type mice. In addition, we provide data in Figure 3 for Referees showing that EVs have an increased ratio of APP-CTFs/FL-APP relative to lysates and that EVs are enriched for mature, glycosylated FL-APP in our cell culture experiments.

11. As for p62 in EV, there are repetitive discussion of the existence of p62 in the EV after VPS34 inhibition in primary cultured neurons and N2a cells (Fig. 4b,d), which suggests the shuttling of dysfunctional endolysosomes to EV as p62-containing complex. However, p62 expression is not shown in the EV fraction isolated from VPS34 cKO mice (Fig. 7d). This is a key conclusive data and should be presented.

We had previously attempted without success to detect p62 in our original two independent EV purification experiments, although we were able to show p62 immunoreactivity in hippocampal lysates, with an increase in the cKO. During the revision, we conducted an additional third experiment (see Figure 5 for Referees). While we were still not able to detect p62, we confirmed for a third time that EVs derived from cKO mice are enriched for APP-CTFs. We conclude that p62 is therefore not enriched on brain exosomes derived from Vps34 cKO mice, contrary to exosomes derived from neurons/N2a cells treated with VPS34IN1. We mention this point in the revised manuscript on page 16 (lines 374-376).

Figure 5. Lack of detection of p62 in EVs derived from mouse brain. EVs were purified from CTRL and cKO mice as described in Method sections and processed for Western Blot analysis.

12. In Fig. 6b, treatment of cells with nSMase2 inhibitor GW4869 would suppress the total number of EV secreted to the media. Please clarify how much is the input of the EV for each sample.

We thank the reviewer for raising this important point. As mentioned in the Methods sections, cells were treated and the same volume of media from each sample was processed for purification of exosomes. In the particular case of GW4869 and myriocin treatment, 1×10^6 neurons were seeded in 6-wells, matured for 15-18 days and incubated in 1.5mL of media during drug treatment. The 100K pellets were resuspended in equal volumes of RIPA buffer and the volume corresponding to approx. 75% of all pelleted material was loaded on SDS-PAGE. Considering we were loading an equivalent volume per sample/cell mass, we normalized the levels of all proteins to the protein concentration of the cell lysate. We clarified this point in the revised Methods section on page 28 (lines 667-669).

13. This reviewer disagrees with the conclusion that Atg5 KO has no effect on the secretion of APP-CTF in the EV fraction. Supplementary Fig. 6d shows enhanced accumulation of APP-CTF in ATG5 KO N2a cells, which appears to be more than wildtype N2a cells (lane 1) and is further accumulated by VPS34IN1 treatment (lane 4). This clearly shows that ATG5 deletion inhibit autophagic function and enhances the secretion of APP-CTF in the EV fraction. The authors show the intensity as fold increase of untreated cells, but this is not the correct analysis of band intensity. The band intensity should be quantified on the same SDS-PAGE after loading the same amount of protein from different cell types. The data presentation and interpretation should be thoroughly revised.

We thank the reviewer for raising this important point and agree in part with his/her assessment. However, while we have updated Supplementary Figure 6 according to the reviewer's recommendation and state that Atg5 KO increases the secretion of APP-CTFs in exosomes, we are still confident regarding our original interpretation, namely that APP-CTF sorting to EVs upon VPS34 inhibition is independent of its role in autophagy impairment. We have updated the text accordingly as to clarify this particular concern on page 13 (line 301-315) of the revised manuscript.

14. The Supplementary Fig. 4c does not have the methods to understand how BMP is stained for imaging, and does not appear to be co-localized with LAMP-1. This description (line 213-215) should be revised or replace the data with more representative image.

We apologize for this omission and for not being clearer in regards to the colocalization between LAMP-1 and BMP. First, we did not mention a specific protocol for BMP staining

because it was performed similarly to the other stainings, as described in the original Methods section. In response to this concern, we now refer the reader to the Methods section in the legend of Supplementary Figure 4c. Secondly, we have rephrased the text on page 10 (line 208-212) to indicate that the BMP immunoreactivity is almost exclusively found in the lumen of LAMP-1-positive compartments rather than co-localizing with LAMP-1, consistent with the notion that it is enriched on intraluminal vesicles (as shown by several other labs (Chevallier et al. 2008; Bissig & Gruenberg 2013; Bache et al. 2003)).

15. *In general, mono-ubiquitination is necessary for the sorting of proteins to ILV via ESCRT machinery, but the protein is deubiquitinated before the insertion into the ILVs. This study repeatedly shows poly-ubiquitinated molecules in the EV fraction (Fig. 4b, 6b, 7d), which is odd but may represent non-ESCRT machinery for the insertion of these molecules to ILVs. Please cite at least one reference reporting the existence of poly-ubiquitinated molecules in the ILVs or exosomes to substantiate the finding.*

We would like to refer the reviewer to two publications reporting and characterizing poly-ubiquitinated proteins in EV fractions (Pisitkun et al. 2004; Huebner et al. 2016). In the first study ubiquitin was detected by LC-MS and immunoblot, spanning a wide molecular mass range from 10kDa to 400kDa, indicating that EVs contained poly-ubiquitinated proteins. The later study reported immunogold labelling of ubiquitin in ILVs and exosomes from human epithelial cells. In addition, they performed LC-MS analysis of poly-ubiquitinated proteins present in exosomes and identified ubiquitin chain in proteins enriched in EVs such as ALIX and TSG101, among others. We are now citing these studies in the revised manuscript in page 11 (line 252-253).

16. *Line 277-278 says "EVs containing BMP can be unambiguously defined as bona fide exosomes". This should be toned down since later in the discussion BMP is described as a unique marker under specific conditions associated with endolysosomal dysfunction (line 471-473).*

We thank the reviewer for pointing out this overstatement, which we have now toned down in the revised manuscript.

Minor comments:

1. *In Fig. 1c image, please clarify if the CatD refers to only processed CatD or both CatD and proCatD.*

We apologize for omitting this information. We have updated Figure 1c mentioning proCatD/CatD.

References

- Aits, S. et al., 2015. Sensitive detection of lysosomal membrane permeabilization by lysosomal galectin puncta assay. *Autophagy*, (August), pp.00–00.
- Bache, K.G. et al., 2003. Hrs regulates multivesicular body formation via ESCRT recruitment to endosomes. *Journal of Cell Biology*, 162(3), pp.435–442.
- Bischoff, V. et al., 2012. Seizure-Induced Neuronal Death Is Suppressed in the Absence of the Endogenous Lectin Galectin-1. *Journal of Neuroscience*, 32(44), pp.15590–15600.
- Bissig, C. & Gruenberg, J., 2013. Lipid sorting and multivesicular endosome biogenesis. *Cold Spring Harbor Perspectives in Biology*, 5(10), p.a016816.
- Chevallier, J. et al., 2008. Lysobisphosphatidic Acid Controls Endosomal Cholesterol Levels. *Journal of Biological Chemistry*, 283(41), pp.27871–27880.
- Cossec, J.-C. et al., 2012. Trisomy for synaptojanin1 in Down syndrome is functionally linked to the enlargement of early endosomes. *Human molecular genetics*, 21(14), pp.3156–72.
- Dall'Armi, C., Devereaux, K. a & Di Paolo, G., 2013. The role of lipids in the control of autophagy. *Current biology : CB*, 23(1), pp.R33-45.

- Devereaux, K. et al., 2013. Regulation of Mammalian Autophagy by Class II and III PI 3-Kinases through PI3P Synthesis. *PLoS ONE*, 8(10), pp.10–12.
- Futter, C.E. et al., 2001. Human VPS34 is required for internal vesicle formation within multivesicular endosomes. *Journal of Cell Biology*, 155(7), pp.1251–1263.
- Greening, D.W. et al., 2015. A Protocol for Exosome Isolation and Characterization: Evaluation of Ultracentrifugation, Density-Gradient Separation, and Immunoaffinity Capture Methods. In *Methods in molecular biology (Clifton, N.J.)*. pp. 179–209.
- Guix, F.X. et al., 2017. Tetraspanin 6: a pivotal protein of the multiple vesicular body determining exosome release and lysosomal degradation of amyloid precursor protein fragments. *Molecular Neurodegeneration*, 12(1), p.25.
- Haass, C. et al., 2012. Trafficking and proteolytic processing of APP. *Cold Spring Harbor perspectives in medicine*, 2(5), p.a006270.
- Heneka, M.T., Kummer, M.P. & Latz, E., 2014. Innate immune activation in neurodegenerative disease. *Nature reviews. Immunology*, 14(7), pp.463–77.
- Huebner, A.R. et al., 2016. Deubiquitylation of Protein Cargo Is Not an Essential Step in Exosome Formation. *Molecular & cellular proteomics : MCP*, 15(5), pp.1556–71.
- Jiang, Y. et al., 2010. Alzheimer's-related endosome dysfunction in Down syndrome is A β -independent but requires APP and is reversed by BACE-1 inhibition. *Proceedings of the National Academy of Sciences*, 107(4), pp.1630–1635.
- Kishi-Itakura, C. et al., 2014. Ultrastructural analysis of autophagosome organization using mammalian autophagy-deficient cells. *Journal of Cell Science*, 127(22), pp.4984–4984.
- Klingeborn, M. et al., 2017. Directional Exosome Proteomes Reflect Polarity-Specific Functions in Retinal Pigmented Epithelium Monolayers. *Scientific Reports*, 7(1), p.4901.
- Kowal, J. et al., 2016. Proteomic comparison defines novel markers to characterize heterogeneous populations of extracellular vesicle subtypes. *Proceedings of the National Academy of Sciences*, 113(8), pp.E968–E977.
- Maejima, I. et al., 2013. Autophagy sequesters damaged lysosomes to control lysosomal biogenesis and kidney injury. *The EMBO Journal*, 32, pp.2336–2347.
- Morel, E. et al., 2013. Phosphatidylinositol-3-phosphate regulates sorting and processing of amyloid precursor protein through the endosomal system. *Nature communications*, 4, p.2250.
- Nemazanyy, I. et al., 2013. Defects of Vps15 in skeletal muscles lead to autophagic vacuolar myopathy and lysosomal disease. *EMBO molecular medicine*, 5(6), pp.870–90.
- Papadopoulos, C. et al., 2017. VCP/p97 cooperates with YOD1, UBXD1 and PLAA to drive clearance of ruptured lysosomes by autophagy. *The EMBO Journal*, 36(2), pp.135–150.
- Paz, I. et al., 2010. Galectin-3, a marker for vacuole lysis by invasive pathogens. *Cellular Microbiology*, 12(February), pp.530–544.
- Perez-Gonzalez, R. et al., 2012. The exosome secretory pathway transports amyloid precursor protein carboxyl-terminal fragments from the cell into the brain extracellular space. *Journal of Biological Chemistry*, 287(51), pp.43108–43115.
- Pisitkun, T., Shen, R.-F. & Knepper, M.A., 2004. Identification and proteomic profiling of exosomes in human urine. *Proc. Natl. Acad. Sci. USA*, 101(36), pp.13368–13373.
- Raiborg, C., Schink, K.O. & Stenmark, H., 2013. Class III phosphatidylinositol 3-kinase and its catalytic product PtdIns3P in regulation of endocytic membrane traffic. *The FEBS journal*, 280(12), pp.2730–42.
- Schink, K.O., Raiborg, C. & Stenmark, H., 2013. Phosphatidylinositol 3-phosphate, a lipid that regulates membrane dynamics, protein sorting and cell signalling. *BioEssays*, 35(10), pp.900–912.
- Settembre, C. et al., 2013. Signals from the lysosome: a control centre for cellular clearance and energy metabolism. *Nature reviews. Molecular cell biology*, 14(5), pp.283–296.
- Sharples, R.A. et al., 2008. Inhibition of gamma-secretase causes increased secretion of amyloid precursor protein C-terminal fragments in association with exosomes. *FASEB journal : official*

- publication of the Federation of American Societies for Experimental Biology*, 22(5), pp.1469–78.
- Théry, C. et al., 2006. Isolation and Characterization of Exosomes from Cell Culture Supernatants. *Current protocols in cell biology / editorial board, Juan S. Bonifacino ... [et al.]*, Chapter 3, pp.1–29.
- Wang, L., Budolfson, K. & Wang, F., 2011. Pik3c3 deletion in pyramidal neurons results in loss of synapses, extensive gliosis and progressive neurodegeneration. *Neuroscience*, 172, pp.427–442.
- Williamson, R.L. et al., 2017. Disruption of amyloid precursor protein ubiquitination selectively increases amyloid beta (A β) 40 levels via presenilin 2-mediated cleavage. *Journal of Biological Chemistry*, p.jbc.M117.818138.
- Zhang, Y. et al., 2014. An RNA-sequencing transcriptome and splicing database of glia, neurons, and vascular cells of the cerebral cortex. *The Journal of neuroscience : the official journal of the Society for Neuroscience*, 34(36), pp.11929–47.

REVIEWERS' COMMENTS:

Reviewer #1 (Remarks to the Author):

I am satisfied with the answers to the concerns raised in my initial review

Reviewer #2 (Remarks to the Author):

The authors have submitted a revised version of their manuscript demonstrating that APP CTF fragments are shunted to a population of EVs upon vps34 inhibition-induced lysosomal dysfunction. After reading their full rebuttal and revised data, I have to conclude that, overall, the authors have significantly improved the manuscript and have addressed my major concerns. The manuscript is now more clearly written out so that the main message is better advocated. In principle I can agree with publication, but have still a few outstanding issues that the authors should consider.

I'm still struggling with the interpretation and identity of the p62-/flotillin-positive organelles. I can agree that the enlarged endosomal structures might be the lamp1-positive organelles, and maybe this should be as such also mentioned in the text (line 184-188). If the dark organelles are suggested to be the p62-organelles, then they are clearly separate organelles. However, the way the authors describe their linescans of the superresolved images remains ambiguous by stating that '... were generally distributed to the periphery of lamp1-positive organelles': it is not clear whether they mean on the periphery of lamp1-positive organelles or clearly a separate organelle closely apposed to endolysosomes.

They now more clearly state that the p62-positive organelles are most likely earlier stages of endosomes on which I can agree: having said that, and given the fact that these organelles colocalize with gal3 (and are thus damaged), wouldn't that more agree with the definition of amphisomes or the direct recruitment of the autophagy-machinery to damaged endosomes (as seen in other specific cases of organelle damage like mitophagy, ER-phagy)? To my opinion that would explain the ambiguity on whether autophagy is included or not as this is a specific case of targeting damaged endosomes. I would suggest that the authors should consider incorporating this alternative explanation of their observations and on this population.

Small remark:

- A better contrasted picture for the Lamp1 staining in suppl fig 4C, vps34inh should be provided. It is not so clear (compared to WT) that BMP is in all these lamp1 positive organelles because of the weak signal.

- In the attempt to improve contrast of the Rab5 immunostaining (figure 1a), the increased intensity of rab5 positive structures in VPS34IN1 is completely gone, not only in intensity in individual spots but also in the number of spots (control is far higher compared to the VPS34IN1). Authors should look back at this and find kind of a representative image.

- With respect to the ATG5KO data, the authors now state more explicitly that 'secretion of EVs induced by Vps34 inhibition occurs independently of any autophagic defects and originates from other aspects of endosomal dysfunction' (page 13, l 306-307). Although I agree that the observed effects are majorly coming from endolysosomal regulation, they cannot exclude that there is some component of autophagy involved. Sentence should be

changed to "...occurs independently from major autophagic defects...".

Reviewer #3 (Remarks to the Author):

The major concerns are all addressed in the revised manuscript and it was really improved in terms of data presentation, description and discussion with adequate citations.

RESPONSE TO REVIEWERS' COMMENTS:

Reviewer #2 (Remarks to the Author):

The authors have submitted a revised version of their manuscript demonstrating that APP CTF fragments are shunted to a population of EVs upon vps34 inhibition-induced lysosomal dysfunction. After reading their full rebuttal and revised data, I have to conclude that, overall, the authors have significantly improved the manuscript and have addressed my major concerns. The manuscript is now more clearly written out so that the main message is better advocated. In principle I can agree with publication, but have still a few outstanding issues that the authors should consider.

I'm still struggling with the interpretation and identity of the p62-/flotillin-positive organelles. I can agree that the enlarged endosomal structures might be the lamp1-positive organelles, and maybe this should be as such also mentioned in the text (line 184-188). If the dark organelles are suggested to be the p62-organelles, then they are clearly separate organelles. However, the way the authors describe their linescans of the superresolved images remains ambiguous by stating that '... were generally distributed to the periphery of lamp1-positive organelles': it is not clear whether they mean on the periphery of lamp1-positive organelles or clearly a separate organelle closely apposed to endolysosomes.

We thank the reviewer for requesting a better clarification of this important point. We believe that p62/flotillin-2 positive structures are distinct organelles apposed to LAMP-1 compartments based on the distinct punctate morphology in confocal images and presence of separate electron-dense organelles in electron microscopy. Following the reviewer's request, we have updated the text to clarify this ambiguity (page 9, line 223-227 and page 10, line 266-270).

They now more clearly state that the p62-positive organelles are most likely earlier stages of endosomes on which I can agree: having said that, and given the fact that these organelles colocalize with gal3 (and are thus damaged), wouldn't that more agree with the definition of amphisomes or the direct recruitment of the autophagy-machinery to damaged endosomes (as seen in other specific cases of organelle damage like mitophagy, ER-phagy)? To my opinion that would explain the ambiguity on whether autophagy is included or not as this is a specific case of targeting damaged endosomes. I would suggest that the authors should consider incorporating this alternative explanation of their observations and on this population.

We have updated our manuscript (page 10, line 279) to mention that these damaged organelles are marked for degradation via recruitment of autophagy adapter p62, however lack of PI3P-dependent LC3 lipidation and autophagosome elongation likely prevent their efficient clearance. We have also mentioned another scenario, whereby those damaged structures could be late endosomes or lysosomes that have lost their membrane markers through proteolytic degradation occurring upon loss of membrane integrity (page 11, line 335-337).

Small remark:

- A better contrasted picture for the Lamp1 staining in suppl fig 4C, vps34inh should be provided. It is not so clear (compared to WT) that BMP is in all these lamp1 positive organelles because of the weak signal.

We have updated Supplementary Figure 4C accordingly.

- In the attempt to improve contrast of the Rab5 immunostaining (figure 1a), the increased intensity of rab5 positive structures in VPS34IN1 is completely gone, not only in intensity in individual spots but also in the number of spots (control is far higher compared to the VPS34IN1). Authors should look back at this and find kind of a representative image.

We have updated Figure 1A accordingly with the inclusion of a new set of figures, including super-resolution Airy Scan insets where endosomal enlargement and intensity difference is

now more clear. We also took the opportunity to select a better inset in Fig. 1c that better reflects the luminal localization of CatD after VPS34IN1 treatment.

- With respect to the ATG5KO data, the authors now state more explicitly that 'secretion of EVs induced by Vps34 inhibition occurs independently of any autophagic defects and originates from other aspects of endosomal dysfunction' (page 13, l 306-307). Although I agree that the observe effects are majorly coming from endolysosomal regulation, they cannot exclude that there is some component of autophagy involved. Sentence should be changed to '...occurs independently from major autophagic defects....'.

We agree with the reviewer and have now toned down our original statement as suggested (page 14, line 445).